# A non-methanogenic archaeon within the order *Methanocellales*

Shino Suzuki [1,2,3,4] ✉, Shun'ichi Ishii [4] ✉, Grayson L. Chadwick[5], Yugo Tanaka[6], Atsushi Kouzuma [6], Kazuya Watanabe[6], Fumio Inagaki [7,8], Mads Albertsen [9], Per H. Nielsen [9] & Kenneth H. Nealson[10]

Serpentinization, a geochemical process found on modern and ancient Earth, provides an ultra-reducing environment that can support microbial methanogenesis and acetogenesis. Several groups of archaea, such as the order *Methanocellales*, are characterized by their ability to produce methane. Here, we generate metagenomic sequences from serpentinized springs in The Cedars, California, and construct a circularized metagenome-assembled genome of a *Methanocellales* archaeon, termed Met12, that lacks essential methanogenesis genes. The genome includes genes for an acetyl-CoA pathway, but lacks genes encoding methanogenesis enzymes such as methyl-coenzyme M reductase, heterodisulfide reductases and hydrogenases. In situ transcriptomic analyses reveal high expression of a multi-heme $c$-type cytochrome, and heterologous expression of this protein in a model bacterium demonstrates that it is capable of accepting electrons. Our results suggest that Met12, within the order *Methanocellales*, is not a methanogen but a $CO_2$-reducing, electron-fueled acetogen without electron bifurcation.

Serpentinization, the process whereby rocks rich in olivine and pyroxene (common components of mantle rocks) react with water, yields magnetite, hydroxide, and serpentine minerals[1]. Additionally, it generates molecular hydrogen, which serves as an energy resource for various chemosynthetic organisms. These environments are recognized as important analogs to potential ancient ecosystems, both on Earth and Mars, where prevalent highly reducing mineral compositions likely characterized the undifferentiated crust[2]. This molecular hydrogen can abiotically reduce carbon dioxide, producing simple organic molecules such as methane, formate and possibly acetate[3–5]. Since these abiotic reactions resemble the core bioenergetic reactions of hydrogenotrophic methanogenesis ($CO_2 + 4\,H_2 \rightarrow CH_4 + 2\,H_2O$) and

acetogenesis ($2\,CO_2 + 4H_2 \rightarrow CH_3COOH + 2H_2O$) via acetyl-CoA pathway, a serpentinization setting has been implicated as a place where biochemical reactions might occur and hence significant for the search for such processes in extraterrestrial systems[1,2,6–8]. Methanogenesis and acetogenesis coupled to the acetyl-CoA pathway are the most ancient pathways, in part because it is the only one that occurs in both domains Bacteria and Archaea and functions in both carbon and energy metabolism[9,10].

Due to the importance in understanding the early or adaptive evolution for life, the microbial diversity of present-day terrestrial and marine serpentinization sites have begun to be described[11–16]. Despite abundant energy sources, serpentinization sites host low cell density

[1]Geobiology and Astrobiology Laboratory, RIKEN Cluster for Pioneering Research, Wako, Saitama, Japan. [2]Institute of Space and Astronautical Science (ISAS), Japan Aerospace Exploration Agency (JAXA), Sagamihara, Kanagawa, Japan. [3]School of Physical Sciences, SOKENDAI (Graduate University for Advanced Studies), Sagamihara, Kanagawa, Japan. [4]Institute for Extra-cutting-edge Science and Technology Avant-garde Research (X-star), Japan Agency for Marine and Earth Science and Technology (JAMSTEC), Yokosuka, Kanagawa, Japan. [5]Department of Molecular and Cell Biology, University of California, Berkeley, Berkeley, CA, USA. [6]School of Life Sciences, Tokyo University of Pharmacy and Life Sciences, Hachioji, Tokyo, Japan. [7]Advanced Institute for Marine Ecosystem Change (WPI-AIMEC), JAMSTEC, Yokohama, Kanagawa, Japan. [8]Department of Earth Science, Graduate School of Science, Tohoku University, Sendai, Miyagi, Japan. [9]Center for Microbial Communities, Aalborg University, Aalborg, Denmark. [10]Department of Earth Sciences, University of Southern California, Los Angeles, CA, USA. ✉e-mail: shino.suzuki@riken.jp; sishii@jamstec.go.jp

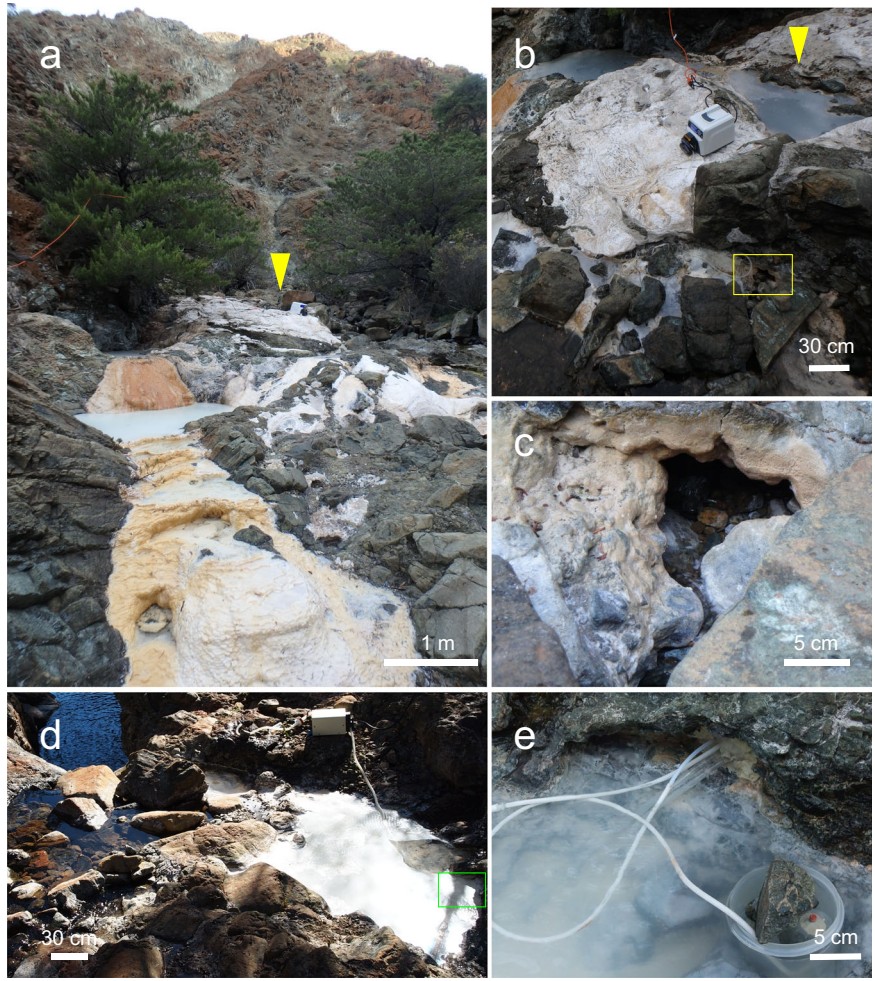

**Fig. 1 | The Cedars serpentinization site and in situ filtration. a** Barnes Spring complex in The Cedars area. Yellow arrowhead shows the location of Barnes Spring 5 (BS5). **b** Bird's eye view of the BS5 pool. **c** The source water of BS5 flow (BS5sc) at the bottom of BS5 pool (yellow rectangle in **b**). **d** Grotto Pool Spring1 (GPS1) located near the Grotto pool near camp site. **e** Microbes in the GPS1 source spring water (sample GPS1) was collected by using in-line filtration system (green rectangle in panel d).

communities, usually attributed to the polyextreme conditions with hyperalkaline pH, low oxidants and limitation of organic carbons, dissolved inorganic carbon (DIC) and phosphate. While presence of acetogens and methanogens are indicated[11,14,17–19], it has not been fully described how this $H_2$-rich but bioavailable $CO_2$-limited ultra-reducing serpentinizing setting can drive acetogenesis and methanogenesis.

The Cedars is a zone of active serpentinization located within the Franciscan Subduction Complex in Northern California[20,21]. The deep groundwaters, delivered from ultramafic rock layers without mixing of surface oxidized water, are highly alkaline (pH 11.5–12.0) and extremely low redox potential ($E_h$ <−650 mV) and host-microbial communities with low cell density ($10^{2.3}$ cells/mL)[21,22]. The deep groundwater contains approximately 680 μM hydrogen, 34 μM methane[23], 7 μM formate, and 70 μM of acetate[14]. The diversity of microorganisms based on metagenome-assembled genomes (MAGs) of three representative sites, the source water of Grotto Pool Spring1 (GPS1), the source water of Barnes Spring 5 (BS5sc), and the Barnes Spring 5 Pool (BS5pool) (Fig. 1), confirmed that the community compositions are stable over seven years. The community composition of the deep groundwater in The Cedars comprises more than 60% are bacterial members of the episymbiotic group Patescibacteria (formally Parcubacteria or candidate division OD1) with the remainder comprised of potential acetogens affiliated with phyla Chloroflexota (formally Chloroflexi), Bacillota (formally Firmicutes), *Candidatus*

Horikoshibacteria (NPL-UPA2) and potential methanogens affiliated with archaeal phyla Halobacteriota (formally Euryarchaeota) and Methanobacteriota (formally Euryarchaeota) (Supplementary Fig. 1a)[11,19]. From the BS5sc metagenome, Met12 in the phylum Halobacteriota is a 6th dominant microorganism (relative frequency of 4.5% based on RpsC gene), and most dominant Archaea in the deep groundwater at The Cedars (Supplementary Fig. 1b).

Here we present findings on the circularized genome of Met12, an archaeon belonging to the Methanocellales order, discovered in serpentinized springs located in The Cedars, California. This discovery marks the first instance within the seven traditional methanogen orders (*Methanobacteriales, Methanococcales, Methanomicrobiales, Methanosarcinales, Methanocellales, Methanopyrales*, and *Methanomassiliicoccales* according to the NCBI taxonomy) where an organism possesses an acetyl-CoA pathway but lacks essential genes for methanogenesis. Through a combination of genomic analysis, in situ transcriptomic studies, and synthetic biology approaches, we propose a novel acetogenic metabolism, presumably stemming from adaptive evolutionary processes.

## Results

### Comparative genomics of the circularized MAG of Met12
We recovered five MAGs of Met12 from five different metagenomic data collected from three springs in two different years and confirmed

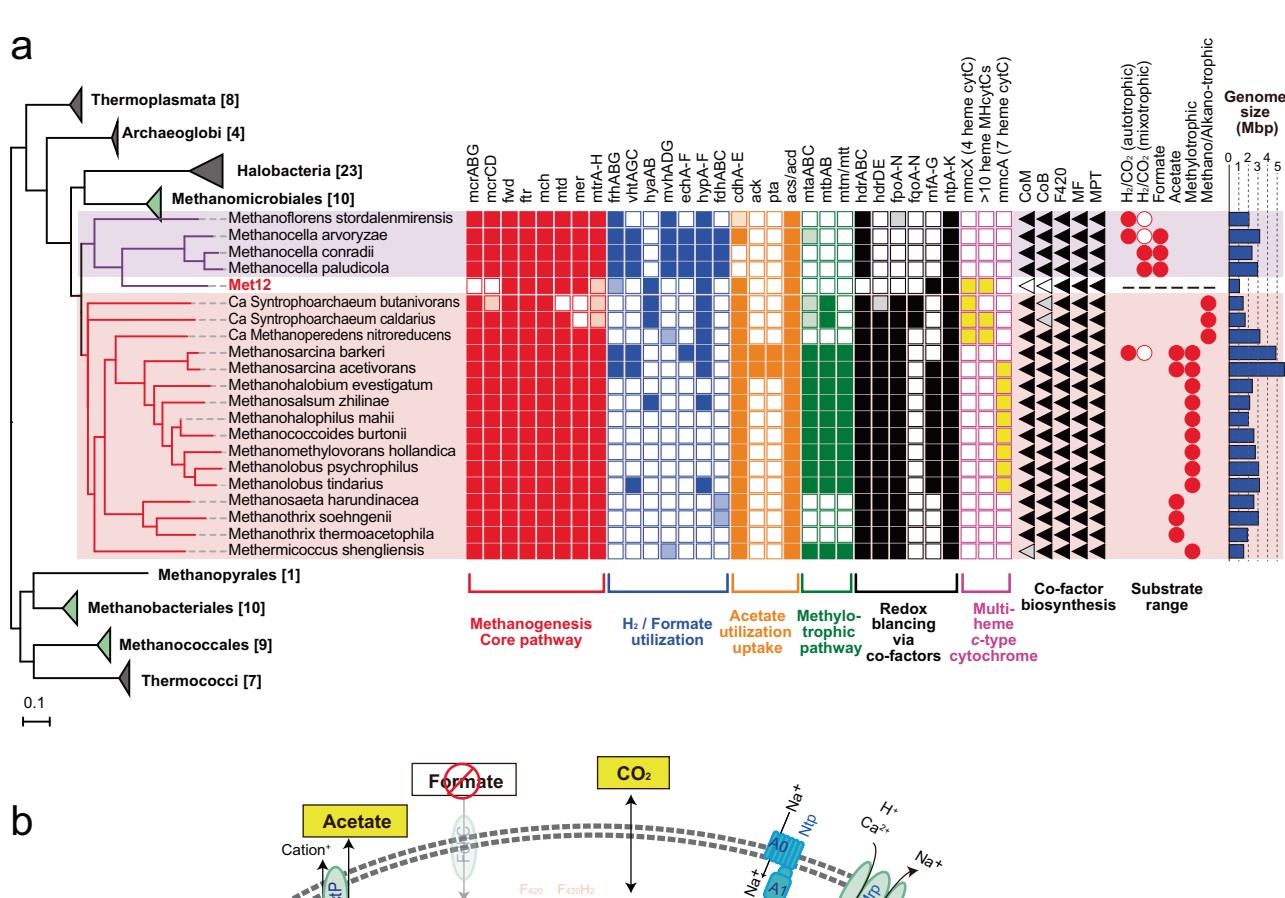

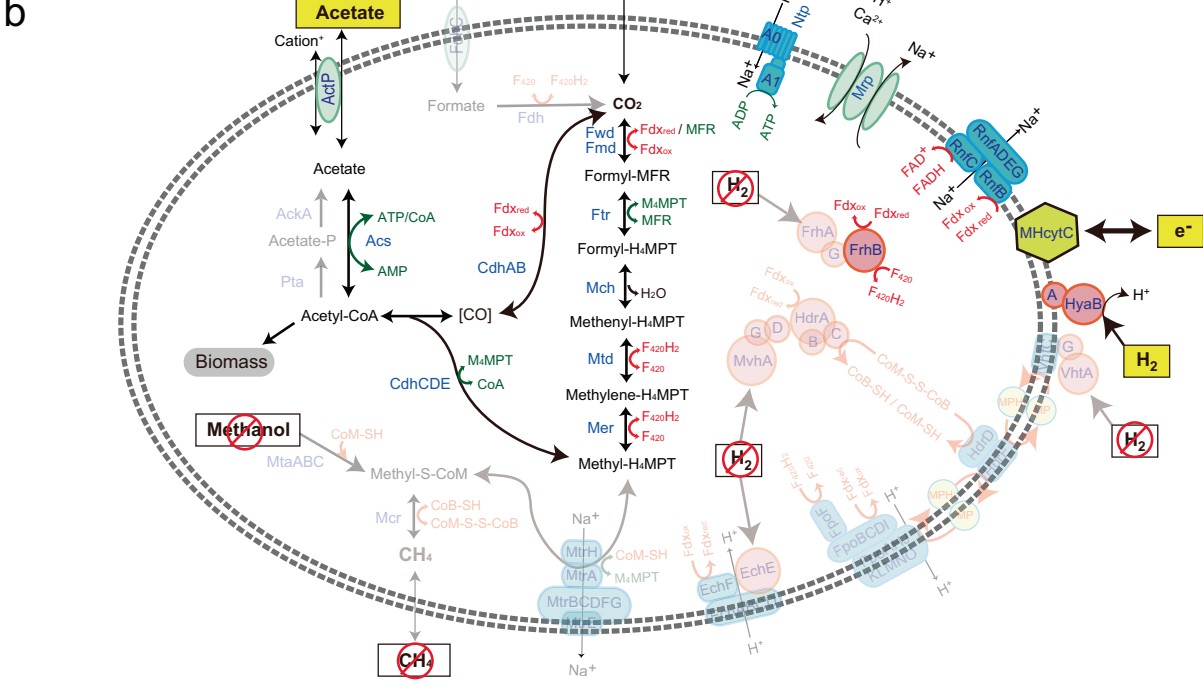

that all the MAGs of the Met12 are nearly identical (Digital DNA-DNA hybridization similarity > 96%) (Supplementary Data 1). To better understand the phylogeny and metabolism of Met12, we further manually refined the MAG by using additional mate-pair information and obtained the circularized MAG of Met12 (Supplementary Data 2, see Supplementary Note 1 for analyzing genome completeness). The circularized genome size was 1,067,436 bp, which is the smallest within all 193 complete genomes in the seven orders of traditional methanogens. The Met12 belongs to the order *Methanocellales*, but further discussed in Supplementary Note 2.

Remarkably, the circularized MAG of Met12 does not encode the methyl-coenzyme M reductase (McrABG) (Fig. 2a, Supplementary

Fig. 2, Supplementary Data 3), the crucial enzyme catalyzing methane production in methanogens or consumption in the anaerobic methane oxidizers (ANME) that use the enzyme in the reverse direction[24]. Since key enzymes for methane production are highly conserved within all seven orders of traditional methanogens, Met12 provides us with the only example of a member in those orders that has been rendered incapable of methane metabolisms (Supplementary Data 3–4). The Met12 further lacks other important genes for methane metabolic pathway, including for the biosynthesis of specific cofactors, CoM, CoB, F430 (ComA-E, AksA-F, CfbA-E) and for the N5-methyl-tetra-hydromethanopterin:coenzyme M methyltransferase complex (MtrABCDEGH) (Fig. 2a). Since these enzymes produce or consume the

**Fig. 2 | Phylogenomic position and key genes for methanogenesis encoded in the *Methanocellales* archaeon Met12. a** Maximum likelihood phylogenetic tree of publicly available genomes belonging to a phylum Halobacteriota with newly identified MAG, Met12 (Left). Orders Methanocelles (purple) and Methanosarcinales (red) were highlighted respectively. The existence of key genes for methanogenesis in the genomes is shown in the square and triangle blocks (right). A filled block indicates full gene set of the module coded in the genomes, whereas filled block with lighter color indicates only partial gene set of the module coded in the genomes. The orthologous clusters and the related genes are shown in Supplementary Data 1. Types of energy metabolism determined by physiological analysis of cultivated organisms are shown in red filled circles, whereas those predicted from genomic constitutions are shown in open circles. Genome size is shown by blue bar chart. **b** Energy and carbon metabolisms of the *Methanocellales* archaeon Met12 predicted from the genomic constitution. Proteins, complexes, cofactors and electron carriers are colored based on the functional categories. Cdh acetyl-CoA decarbonylase/synthase complex Fwd/Fmd formylmethanofuran dehydrogenase, Ftr formylmethanofuran:tetrahydromethanopterin N-formyltransferase, Mch methenyltetrahydromethanopterin cyclohydrolase, Mtd methenyltetrahydromethanopterin dehydrogenase, Mer methenyltetrahydromethanopterin reduce,ase, Mtr tetrahydromethanopterin S-methyltransferase, Mcr methyl-coenzyme M reductase, Acs acetyl-CoA synthetase, YaaH succinate-acetate transporter, SbtA $Na^+$-dependent bicarbonate transporter, Frh coenzyme F420-reducing hydrogenase, Ech/Eha/Ehb energy-converting hydrogenase, Mvh F420-non-reducing hydrogenase, Hdr heterodisulfide reductase, Mrp Multiple resistance and pH antiporter, Ntp archaeal type A1AO-ATPase, Rnf proton/sodium-translocating ferredoxin-NAD:oxidoreductase complex, MHcytC multi-heme c-type cytochrome, Fdx ferredoxin, MP methanopterin, MFR methanofuran, $H_4MPT$ tetrahydromethanopterin, CoA coenzyme A, $F_{420}$ coenzyme $F_{420}$. The missing components for the methanogenic/methanotrophic pathway are translucent.

reactants and products of Mcr as well as generate the cofactors required for its function, their absence is consistent with a complete loss of methane metabolism (Fig. 2b, see Supplementary Note 3 for validation of the lacking).

Another notable feature of the Met12 is lack of heterodisulfide reductases (HdrABC, HdrDE) that are enzymes for the reduction of the disulfide of coenzyme M and coenzyme B (CoMS–SCoB) at the final step of methanogenesis. The HdrABC plays additional roles in driving essential endergonic reactions at the expense of the exergonic reduction of CoMS–SCoM by flavin-based electron bifurcation (FBEB)[25,26]: At the first step of the acetyl-CoA pathway, a reduced form of ferredoxin ($Fdx_{red}$) with mid-potential around −500 mV is required for the most endergonic reaction, which produces CO by reducing $CO_2$ with carbon monoxide dehydrogenase (CdhABCDE) or formylmethanofuran by reducing $CO_2$ with formylmethanofuran dehydrogenases (Fwd/Fmd) (Fig. 2b). The reduction of ferredoxin with hydrogen whose mid potential is −430 mV as the reductant is an endergonic reaction, thus ferredoxin is generally reduced by FBEB or membrane-associated energy-converting hydrogenases in methanogens[26–28]. The MAG of Met12 does not encode [NiFe]-hydrogenase (MvhADG) and formate dehydrogenase (FdhABC), both of which interact with HdrABC to transfer electrons from hydrogen or formate, indicating that the Met12 lost Hdr-related FBEB completely. Met12 does not encode membrane-associated energy-converting [NiFe]-hydrogenases (EchA-F, EhaA-T, EhbA-Q, and MbhA-N) which generates $Fdx_{red}$ by using membrane gradient[26,28]. Met12 does encode catalytic subunits of membrane-bound [NiFe]-hydrogenase (HyaAB) but does not encode HyaC (Fig. 2). Orthologous analysis has revealed that HyaABC are orthologues of VhtAGC (Supplementary Data 3). Given that VhtC (HyaC) is responsible for transferring electrons to the methanophenazine in the membrane, this hydrogenase in Met12 is unable to participate in the reduction of Fdx within the cell[29]. Considering lack of FBEB and membrane-associated energy-converting hydrogenases, Met12 is presumably incapable of fixing $CO_2$ coupled to oxidation of hydrogen or formate with known strategies of methanogenesis or acetogenesis.

Met12 also lacks any methyltransferase system capable of utilizing methylated compounds, such as methanol:coenzyme M methyltransferase (MtaABC). Met12 does however encode acetyl-CoA synthetase (Acs/Acd), raising the possibility that acetate production or consumption is part of its energy metabolism. In addition, two genes for carbon species transporters, the acetate transporter (ActP)[30] and the high-affinity bicarbonate transporter (SbtA)[31], were identified, suggesting the potential interaction of Met12 with extracellular acetate and bicarbonate pools. With these unique features that Met12 is not a methanogen and also lost the FBEB (Fig. 2b), we sought to better understand the energy metabolism of Met12.

## In situ transcriptional analysis

To identify the active metabolic pathways of Met12, four independent in situ gene expression profiles were analyzed at GPS1, BS5sc, and BS5pool in multiple years (Fig. 3, Supplementary Data 5). Briefly, GPS1 (Fig. 1d, e) fed solely by a deep groundwater source that interacts with both the peridotite body and the kilometer-deep marine sediments of the Franciscan Subduction Complex, and BS5 (Fig. 1a, b) fed by a mixture of the deep groundwater source and a shallow groundwater source that interacts only with the overlying peridotite body[21]. BS5sc (Fig. 1c) denotes the pristine water from the BS5 source, while BS5pool (Fig. 1b) denotes the water in a small pool supplied by the BS5 source water. This water in the pool is exposed to sunlight and air. Exceptionally high expression was seen in two genes for all four conditions, which were annotated as a multi-heme *c*-type cytochrome (MHcytC) (ATZ61495.1, we named MmcX) and an archaeal pilin (ATZ61557.1). The average mRNA-RPKM/DNA-RPKM value of the four datasets was 10.31 for MmcX and 9.83 for the pilin, both of which were nearly ten times higher than the median value (0.94) of all the genes coded by Met12.

The MmcX has four heme-binding motifs (CXXCH), a signal peptide and a membrane-binding site. Phylogenetic analyses revealed that the MmcX is in the clade of MHcytCs with three to five heme-binding motifs found in the genomes of *Methanoperedenaceae* (ANME2d), *Methanophagales* (ANME-1), ANME2 cluster of *Methanosarcinales*, *Syntropharchaeales* and *Archaeoglobales* (Fig. 4a, Supplementary Fig. 3, Supplementary Data 6). Since the MHcytCs are known to be involved in an extracellular electron transfer (EET) reaction from/to solid mineral or microbial surface and upregulate the gene expression during the EET, it is potentially an EET unit on the cell membrane[32]. The protein structure of MmcX of Met12 deduced by the AlphaFold2 (Fig. 4b, Supplementary Data 7) revealed that the MmcX maintains the closely stacked heme arrangements (≤ 12 Å) (Fig. 4c, d), which is lower than the upper limit of efficient biological electron transfer (<20 Å)[33]. Recently, a homolog of MmcX in *Archaeoglobus veneficus* was reported as an extracellular cytochrome nanowire (AvECN) (AEA46122)[34]. The protein folding structure of the MmcX in Met12 was quite similar to the AvECN (8E5G) as root-mean-square deviation of atomic positions (RMSD) of 3.368 Å, suggesting that the MmcX is also a nanowire.

The other highly expressed gene, pilin, is the main component of pili, that are known to be involved in cell adhesion to solid surface and some of them are indicated to facilitate the electron transfer[35]. The implication that both MmcX and pili are involved in EET from and/or to solid materials (minerals and microbes)[36] suggested that Met12 imports or exports electrons as part of its energy metabolism (Fig. 2b). Further discussion about these two genes and the other highly expressed genes were described in Supplementary Note 4.

## Heterologous expression and characterization of MmcX

To better understand the role of MmcX in the energy metabolism of Met12, MmcX was heterologously expressed in *Shewanella oneidensis*

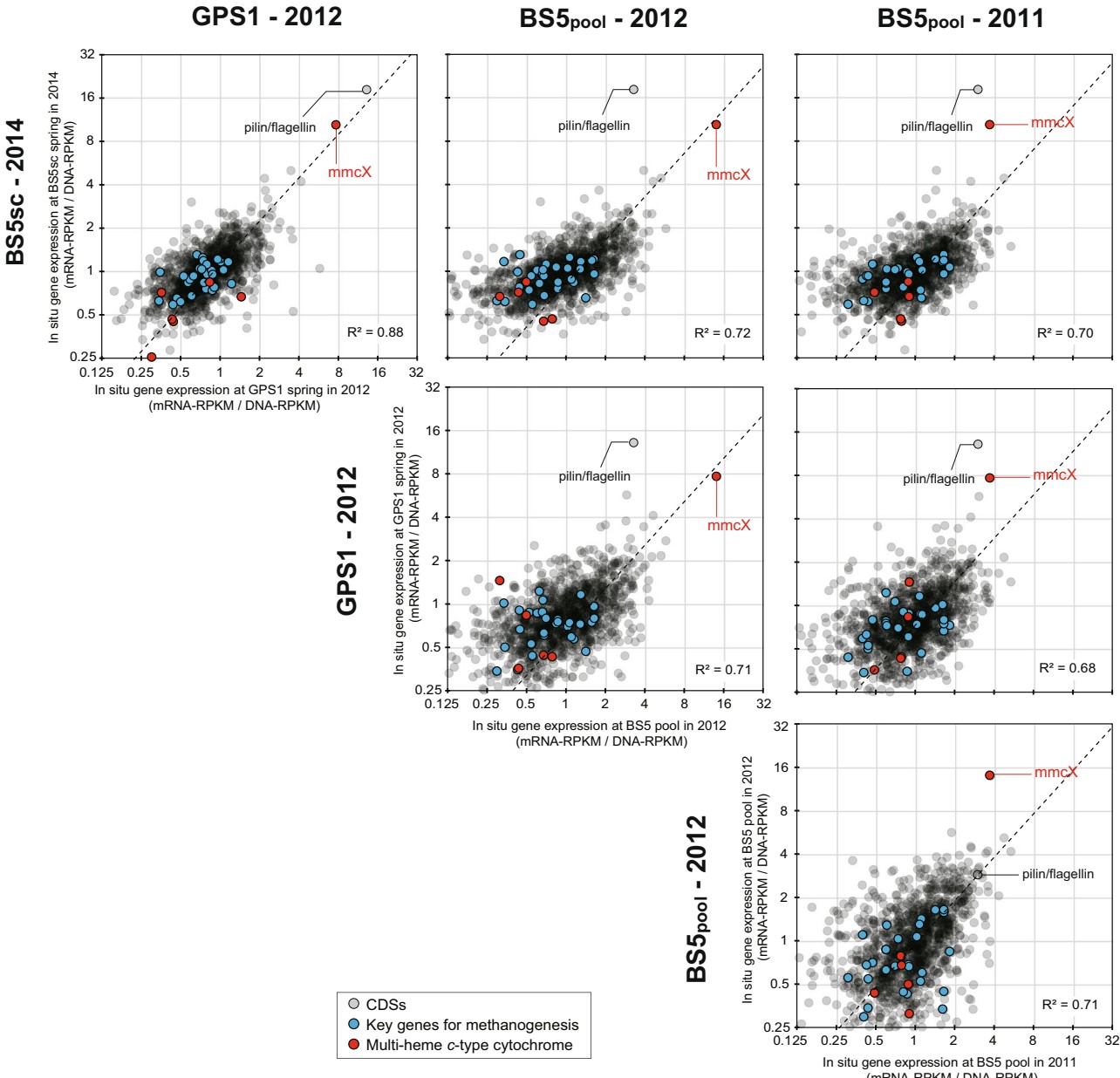

**Fig. 3 | Scatter plots of in situ gene expression for Met12 among BS5sc in 2014, GPS1 in 2012, BS5 pool in 2011, and BS5 pool in 2012.** The scatter plots were generated based on in situ gene expression profiles of the whole CDSs of Met12 (Supplementary Data 2). Proximate line was shown as the linear model of the expression correlation (dashed line), and R-squared value was calculated. Key genes for methanogenesis (Supplementary Data 1) were highlighted as blue circles, while multi-heme c-type cytochromes (MHcytCs) were highlighted as red circles. Highly expressed archaeal pilin and 4-heme MHcytC, *mmcX*, were indicated.

strain MR-1 (Supplementary Fig. 4), a model organism for the study of EET[37]. The strain MR-1 is capable of exporting electrons to an anodic electrode coupled with the oxidation of lactate[38], and also importing electrons from a cathodic electrode coupled with the reduction of fumarate or oxygen for respiration[39]. In either case, outer membrane deca-heme *c*-type cytochromes, MtrC and OmcA, play essential roles in electron transfer. Therefore, we inserted the *mmcX* gene, adapted to the codon usage for Gammaproteobacteria (*Escherichia coli*), into an electrically deficient MR-1 mutant (Fig. 5a, b) lacking *mtrC* and *omcA* (*ΔomcAΔmtrC*) and investigated the electron transfer activities of MmcX.

To determine the electron importing capacity, wild-type MR-1, *ΔomcAΔmtrC* mutant and *ΔomcAΔmtrC* complemented with MmcX (*mmcX-ΔomcAΔmtrC*) were tested on an electrode poised at −400 mV (vs. standard hydrogen electrode, SHE) coupled with the fumarate

reduction. Notably, *mmcX-ΔomcAΔmtrC* rescued electron uptake (Fig. 5a) and the electric current density of strain *mmcX-ΔomcAΔmtrC* was 2.3 times higher than that of the wild-type MR-1 at twenty hours.

As for the electron exporting capacity, the three strains were examined on an anode poised at +200 mV vs. SHE coupled with lactate oxidation, and confirmed that ΔMtrCΔOmcA mutant had one-fifth the electron exporting capacity of wild-type MR-1. In contrast to electron uptake, *mmcX-ΔomcAΔmtrC* was incapable of rescuing electron transfer, and the electron transfer rate was one-tenth of wild-type MR-1, which is lower than those of *ΔomcAΔmtrC* mutant (Fig. 5b). This electrochemical physiology suggests that MmcX of Met12 has an electron importing capacity, but not exporting within those range of potential. The electron importing capability of MmcX indicated that the energy metabolism of Met12 is EET-fueled acetogenesis in serpentinized springs (Fig. 5c).

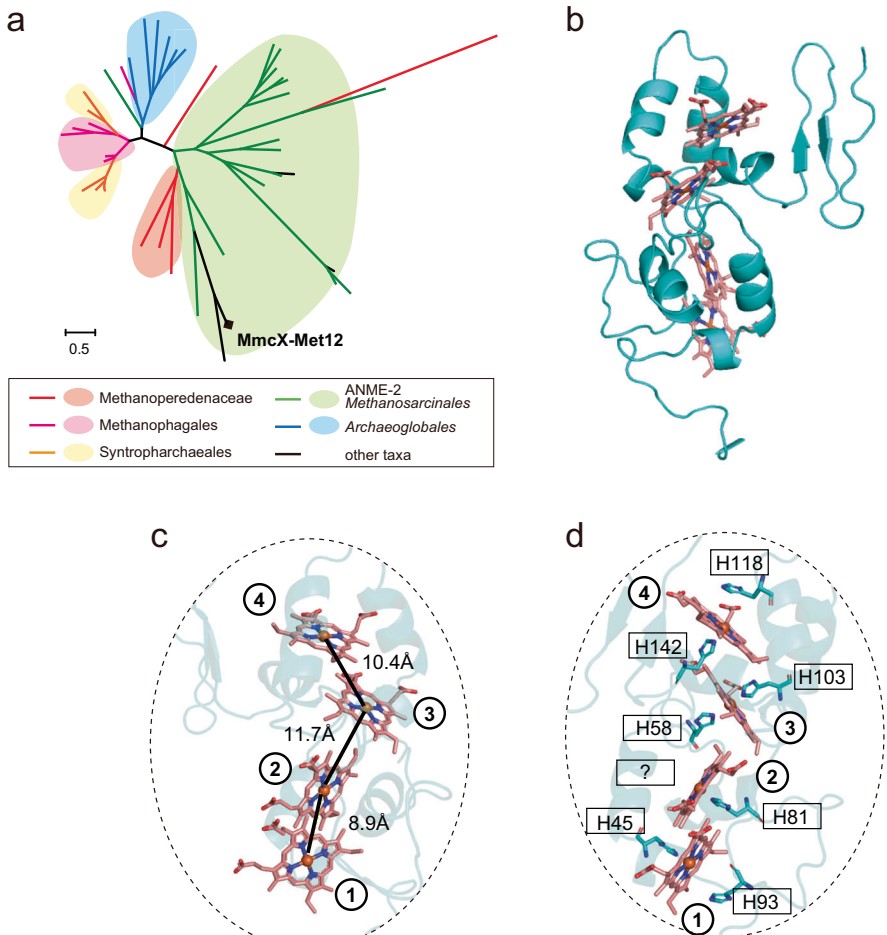

**Fig. 4 | Phylogeny and 3D structure of 4-heme c-type cytochrome MmcX of the Met12. a** Protein sequences of MmcX of the Met12 and the close relatives were aligned using MUSCLE and organized into a phylogenetic tree using maximum likelihood (ML). **b** The 3D protein structure model of MmcX of the Met12. The protein backbone trace was cyan, and the four heme molecules were shown in red

ball and stick. **c**, A close-up view of the MHcytC to show the heme array in MmcX, with the distances between iron atoms of hemes. Heme molecules are labeled from N-terminal with numbers in circles. **d**, The heme array in MmcX of the Met12 and the histidine coordination for the hemes.

## A model for electro-acetogenesis in Met12

A major challenge for any autotrophic metabolism in serpentinizing systems is the acquisition of inorganic carbon. The bicarbonate transporter SbtA in Met12 (ATZ60936.1) showed the highest similarity (amino acid identity = 70%) to those from *Serpentinimonas* strains, facultative chemolithoautotrophs isolated from the BS5 pool of The Cedars, fix carbon from solid calcium carbonate at pH 11[40,41]. Additional MAGs from The Cedars and other serpentinized sites were also found to encode SbtA[12,17] (Fig. 6), suggesting that carbon acquisition from vanishingly small amount of bicarbonate in solution or perhaps solid phase carbonates involving sodium symport is a widespread strategy for carbon uptake in such a DIC-limited serpentinized setting.

Since Met12 is deduced to be an electron-fueled acetogen, the electron source is required for generating $Fdx_{red}$. Common highly reduced minerals observed in the serpentinized setting, like ferran brucite which could have a sufficiently negative potential at pH 12 to reduce $CO_2$[42], is a thermodynamically viable donor for electron transfer to the Met12 via MmcX (Fig. 5c). In addition, membrane-bound HyaAB hydrogenases may also be able to transfer electron from hydrogen to MmcX and further contribute to decrease surrounding pH for importing bicarbonate from calcium carbonate. Since in situ gene expression level of the hydrogenase (0.6) is lower than the median of all genes coded by the Met12 (0.9), the contribution of HyaAB may be limited in The Cedars springs.

In addition to the electron source, its most conspicuous electron-carrying protein complex is a member of the Rnf family[26]. These complexes are understood to play an essential role in the energy metabolism of many bacterial acetogens and archaeal methanogens. All known Rnf family proteins interact with ferredoxin and the electron transfer is coupled to sodium ion translocation, but the electron transfer partner varies. In *Methanosarcina acetivorans*, a membrane-bound MHcytC, MmcA, is the one of the partners of Rnf, and the MmcA is capable of exporting electron to the electron shuttle anthraquinone-2,6-disulfonate (AQDS) coupled with the oxidation of $Fdx_{red}$[43], and also importing electron released from *Geobacter metallireducens* to support the reduction of ferredoxin[44]. MmcX may play a similar role of MmcA, but only for importing electrons in Met12.

The enzymes of the acetyl-CoA pathway require reduced $F_{420}$ in addition to $Fdx_{red}$. While $F_{420}$ is generally reduced by $F_{420}$-reducing hydrogenase (Frh) or $F_{420}$:methanophenazine oxidoreductases (Fpo) in methanogens[28], Met12 encodes only one subunit (FrhB) of Frh (FrhABG) and does not encode an Fpo complex (Fig. 2). The FrhB is homologous to FpoF, which has been characterized as a soluble Fd:$F_{420}$ oxidoreductase in *Methanosarcina*, and as such it is likely that FrhB provides a path for electrons from $Fdx_{red}$ to $F_{420}$ for the two steps of the acetyl-CoA pathway requiring reduced $F_{420}H_2$[45] (Fig. 5c). With $Fdx_{red}$ and $F_{420}H_2$, Met12 can generate acetyl-CoA, and via Acs, produce ATP by substrate-level phosphorylation. To complete the

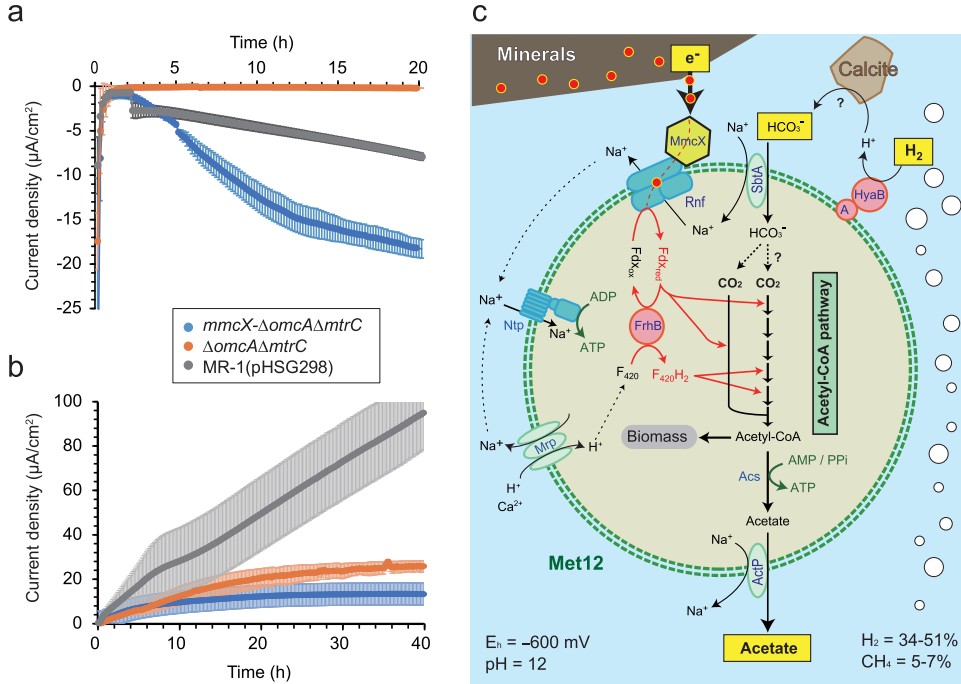

**Fig. 5 | Characterization of 4-heme c-type cytochrome MmcX and reconstruction of metabolic pathway of Met12. a** Electrotrophic current consumption from an electrode poised at −400 mV (vs. SHE) for wild-type MR-1, *ΔomcAΔmtrC* mutant and *mmcX-ΔomcAΔmtrC*. Data are presented as mean values ± SD (n = 3 biologically independent experiments). **b** Electrogenic current generation to an electrode poised at +200 mV (vs. SHE) for wild-type MR-1, *ΔomcAΔomcA* mutant and *mmcX-ΔomcAΔmtrC* mutant. Data are presented as mean values ± SD (n = 3 biologically independent experiments). **c** Estimated electron-fueled acetogenic pathway of the Met12 in highly alkaline ultra-reducing serpentinized subsurface groundwater.

metabolism in the Met12, acetate and sodium must be excluded from the cell[46], through the combined action of the ActP cation/acetate symporter, Rnf, multiple resistance and pH adaptation complexes (Mrp) and Na⁺-dependent ATP synthase coded by the Met12 (Fig. 5c).

### Global distribution of Met12
The 16S rRNA gene-based analysis has indicated a global distribution of Met12-like organisms in serpentinized settings[16,47]. In this study, a total of eight sites from metagenomic datasets were further examined, comprising three oceanic sites and four terrestrial sites, in addition to The Cedars, which identified Met12-like genomes in all oceanic serpentinization sites, Calypso fluids in Lost City hydrothermal field, Mid-Atlantic Ridge[12], ST09 in Prony Bay hydrothermal field, New Caledonia[13], and Old City hydrothermal field in the southwest Indian ridge[16] (Fig. 7). The MAG-838 at the Lost City also suggested the absence of Mcr, associated methanogenesis proteins and Hdr, appearing to be a common feature of these serpentinization inhabitants[12]. Given that the genomic constitution is extensively different from the other genera in the order *Methanocellales* (Fig. 2a), we propose the Met12 and the globally distributed relatives are as a new candidate genus *Candidatus* Serpentinarchaeum spp.

### Evolutionary perspective of methane metabolism in the phylum Halobacteriota
Most of the members in the traditional methanogenic orders are homogeneous in the type of methane metabolism, being composed of either only $CO_2$-reducing or methyl-reducing members. One exception is the phylum Halobacteriota which display all main types of methane metabolisms as well as anaerobic methane or alkane oxidation[48,49]. Gene gain and loss analysis targeted to the methanogenic orders relative to the *Ca.* Serpentinarchaeum sp. Met12 within the phylum Halobacteriota (*Methanocellales*, *Syntropharchaeales*, *Methanotrichales* and *Methanosarcinales*) confirmed high similarities of Met12

and *Syntropharchaeales* (Fig. 8); only the two lost Mcr and Mtr in addition to the loss of Frh, Mvh, Fdh, and Ech, and subsequently occurred a unique genomic evolution independently. Met12 lost the HdrABC, while *Syntropharchaeales* acquired alkyl-coenzyme M reductase (Acr), $F_{420}H_2$:quinone oxidoreductase (Fqo), and lost parts of the acetyl-CoA pathway such as Mer and Mtd. Thus, Met12 and *Syntropharchaeles* presumably have the same ancestor, and the loss of FBEB by HdrABC in the *Ca.* Serpentinarchaeum is probably a result of adaptation to the highly reduced environments commonly observed in the serpentinized sites. The MmcX is an ancestral gene of these orders, although the MmcX is often replaced by the other MHcytCs such as MmcA during the evolutionary process.

### Taxonomic proposal
Description of *Candidatus* Serpentinarchaeum gen. nov. (Ser.pen.ti.n.ar.chae'um). N.L. neut. n. serpentinum a dark green mineral produced from reaction of olivine with water; Latinized G. n. N.L. neut.n. archaeum, an old one/a taxonomic unit; N.L. neut. n. Serpentinarchaeum, an Archaeum from a serpentinizing site.

Description of *Candidatus* Serpentinarchaeum aceticum sp. nov.
L. n. acetum, vinegar; N.L. neut. adj. a.ce.'ticum, related to acetic acid.

### Discussion
Here we report that the dominant Archaea found at The Cedars is an electron-fueled acetogen rather than a methanogen (Fig. 5c), which is a novel metabolic strategy for the traditional methanogenic orders (Fig. 2). As both methanogenesis and acetogenesis have been indicated to be possible in serpentinizing environments, it is of particular interest that Met12's apparent success in adapting to this niche, as evidenced by its persistence in our community analysis and global distribution, involved the loss of methanogenesis. Recent studies reported that *M. acetivorans* C2A that is affiliated with traditional

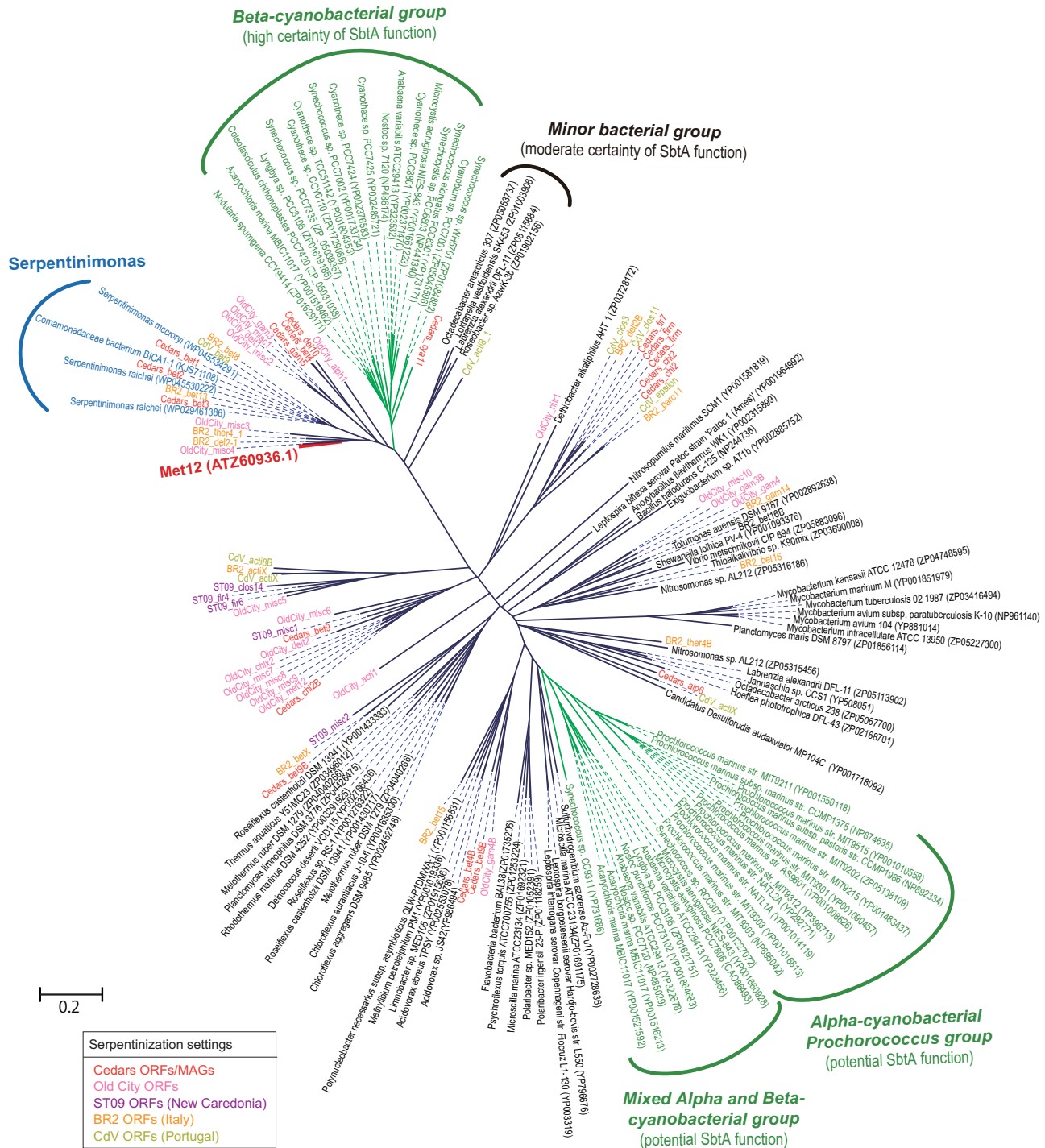

**Fig. 6 | Phylogenetic trees of carbonate alkaline transporter (SbtA) genes.** Protein sequences were aligned using MUSCLE and organized into a phylogenetic tree using maximum likelihood (ML). ORFs from the metagenomes of serpentinization settings are highlighted with different colors. SbtA function was described elsewhere[72].

methanogenic order Methanosarcinales was considered to be obligately methanogenic but can also be converted into an acetogen through autotrophic growth on carbon monoxide and artificial disruption of its methanogenic pathway[50]. Although traditional methanogenic archaea are considered metabolically highly restricted, the experimental evidence of *M. acetivorans* C2A and the in situ environmental evidence of *Ca*. Serpentinarchaeum aceticum Met12 from this study implied that methanogenesis has a capacity to convert to acetogenesis (see Supplementary Note 5 for further discussion).

While the possibility of Met12 being an acetate-oxidizing and iron-reducing archaeon cannot be entirely ruled out, it seems unlikely

considering the results from the heterologous expression and electrochemical experiments of MmcX, as well as the excessively reducing and alkaline environmental conditions which are unsuitable for microbial solid iron (mineral) reduction. Certain homoacetogens can reverse the acetyl-CoA pathway and become acetotrophs, but this process requires a symbiotic partner capable of consuming hydrogen. For instance, *Thermacetogenium phaeum* strain PB is a homoacetogenic bacterium, yet it only oxidizes acetate when cultivated with a symbiotic partner or when a hydrogen concentration is below 0.22 μM[51]. Therefore, acetate oxidation through the reversal of the acetyl-CoA pathway is unlikely to occur because i) potential syntrophic

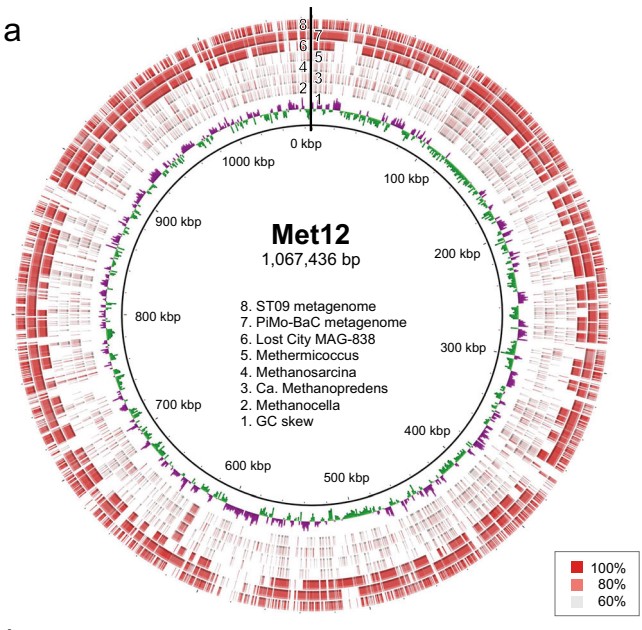

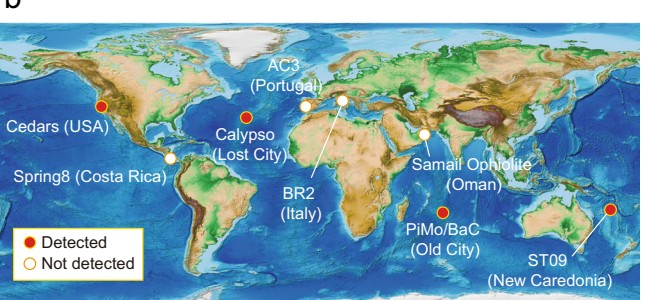

**Fig. 7 | Global distribution of Met12-like MAGs in other serpentinization sites. a** BLAST Ring Image Generator shows the blastn analysis of the Met12 genome against three genomes of close relative strains (*Methanocella paludicola*, *Ca.* Methanoperedens nitroreducens, *Methanosarcina acetivorans*, and *Methermicoccus shengliensis*), MAG-838 from Lost City, the Old City metagenome, and the Prony Bay metagenome. The lower and upper identity threshold in the blastn analysis is shown in the circles. **b** Presence and absence of Met12-like MAGs in various serpentinization sites. The analyses were conducted for The Cedars serpentinizing groundwater (PRJDB2971), Lost City hydrothermal vents (PRJNA779602), Old City hydrothermal vents (PRJNA556392), Voltari Massif travertine, BR2 (PRJNA265986), Cabeco de Vide travertine, AC3 (PRJNA265986), Prony Hydrothermal Field, ST09 (PRJNA265986), and Santa Elena Ophiolite alkaline spring, Spring9 (PRJNA340462). The global map was provided from NOAA National Centers for Environmental Information. 2022: ETOPO 2022 15 Arc-Second Global Relief Model (https://doi.org/10.25921/fd45-gt74. Accessed 2024/03/08)/CC0-1.0.

bacterial partners, such as sulfate, nitrate, or iron reducers, were not identified in the deep groundwater of The Cedars serpentinized site[11], and ii) the serpentinized water in The Cedars is saturated with hydrogen and rich in electrons.

In addition, Met12 presumably utilize an unconventional pathway for the conversion of acetyl-CoA and acetate: unlike most acetogens that employ acetate kinase (AckA) phosphotransacetylase (Pta), Met12 employs Acs. Acs is an atypical enzyme for this process in acetogens, however, it was found that Acs in bacterial acetogen, *Syntrophus aciditrophicus*, and archaeal ANME2d can operate in this direction, supporting energy metabolism by generating ATP from PPi and AMP[52,53]. Interestingly, the genomes of *Ca.* Acetothermia from Samail Ophiolite, recognized as acetogens, lack AckA or Pta but encode Acs[17].

The limited genetic repertoire of Met12's streamlined genome highlights the functions required to survive in these highly reduced

serpentinizing environments. Most metabolic types described from these environments are inferred to utilize reduced chemicals produced as byproducts of the serpentinization reaction. Met12's potential ability to directly uptake electrons implicates it as the first organism presuming to grow directly from the serpentinization reaction itself by physically coupling the oxidation of ultramafic minerals to $CO_2$ reduction via EET.

One of the mysteries is how $CO_2$ fixation might have occurred before proteins enabling electron bifurcation were present[54]. Boyd et al. raised a specific question of how ferredoxin was reduced in ancestral, autotrophic, and anaerobic cells prior to the advent of complex mechanisms[7], and Martin hypothesized that the midpoint potential of hydrothermal effluents stemming from serpentinizing systems can reach −900 mV, which introduces the possibility that organisms living in such environments might not need bifurcation for $Fdx_{red}$ synthesis[9]. While phylogenetic studies based on proteins associated with methanogenesis, such as Mcr, Mtr, Cfb, and Ehb, have suggested that the order Methanocellales is not ancestral[48], the utilization of reduced minerals for the energy metabolism and fixes $CO_2$ without FBEB in the ultra-reduced serpentinized setting may offer an interesting solution[48].

Serpentinization sites are extreme environments on Earth characterized by highly alkaline conditions, limited gradients of oxidation-reduction, and scarcity of phosphorus and bicarbonate. To adapt to these settings, *Ca.* Serpentinarchaeum underwent genome reduction and developed efficient metabolic systems by losing pathways for methane production, which would typically be well-preserved in this lineage. The findings should provide insight into the diversification of carbon fixation capabilities and evolution of carbon fixation among methanogens, as well as the archaeal cellular-level adaptive evolution accompanied by genome reduction.

## Methods
### DNA and RNA Sequencing
Microbial samples were collected from two different low-$E_h$ high-pH springs in The Cedars serpentinization site, Barnes Spring 5 (BS5) (elevation 282 m, N: 38° 37.282′, W: 123° 7.987′) and Grotto Pool Spring1 (GPS1) (elevation 273 m, N: 38° 37.268′ W: 123° 8.014′), by using 0.22 µm in-line filters (Millipore) in 2011 and 2012[11]. In addition to the pool water of the BS5 spring, the source water of the BS5 flow (BS5sc) at the bottom of the BS5 pool (Fig. 1) was also collected in 2014[19]. The filtered cells were immediately frozen with dry ice at the sampling site and stored in −80 °C. Both DNA and RNA were coextracted using a MObio PowerBiofilm RNA Isolation Kit (MO BIO, San Diego, CA, USA) and separated into DNA and RNA using AllPrep DNA/RNA Mini Kit (Qiagen, Germantown, MD, USA)[11,19]. Total RNAs from the samples were treated with a Turbo DNA free kit (Thermo Fisher Scientific, Waltham, MA, USA) for the complete removal of contaminating DNA[18].

A paired-end DNA library of the GPS1, BS5sc, BS5pool, and samples was prepared and sequenced by using an Illumina HiSeq platform as the 101 bp PE or 151 bp PE[11,19]. The DNA sequences have already been deposited in the NCBI Short Read Archive (SRA) under accession numbers DRX086599-DRX086602 (GPS1 and BS5) within BioProject PRJDB2971, and SRX5014375 (BS5sc). In this study, the mate-pair (MP) DNA library of BS5 in 2012 was further sequenced for upgrading the quality of MAGs. The MP DNA library was prepared by using the Illumina Mate Pair kit (Illumina, USA) with the size of DNA fragment of approx. 2 kbp and sequenced by using Illumina HiSeq platform with a 2 × 151 bp run. The raw DNA sequence reads have been deposited in the NCBI SRA under accession number SRX17443947.

Total RNA samples for GPS1 in 2011, BS5 in 2011 and 2012, and BS5sc in 2014 were directly applied for library construction by using ScriptSeq v2 (Illumina, San Diego, CA, United States) without rRNA removal step to avoid unnecessary bias. The RNA libraries were sequenced using Illumina HiSeq2000 platform (Illumina, San Diego, CA, United States) as the

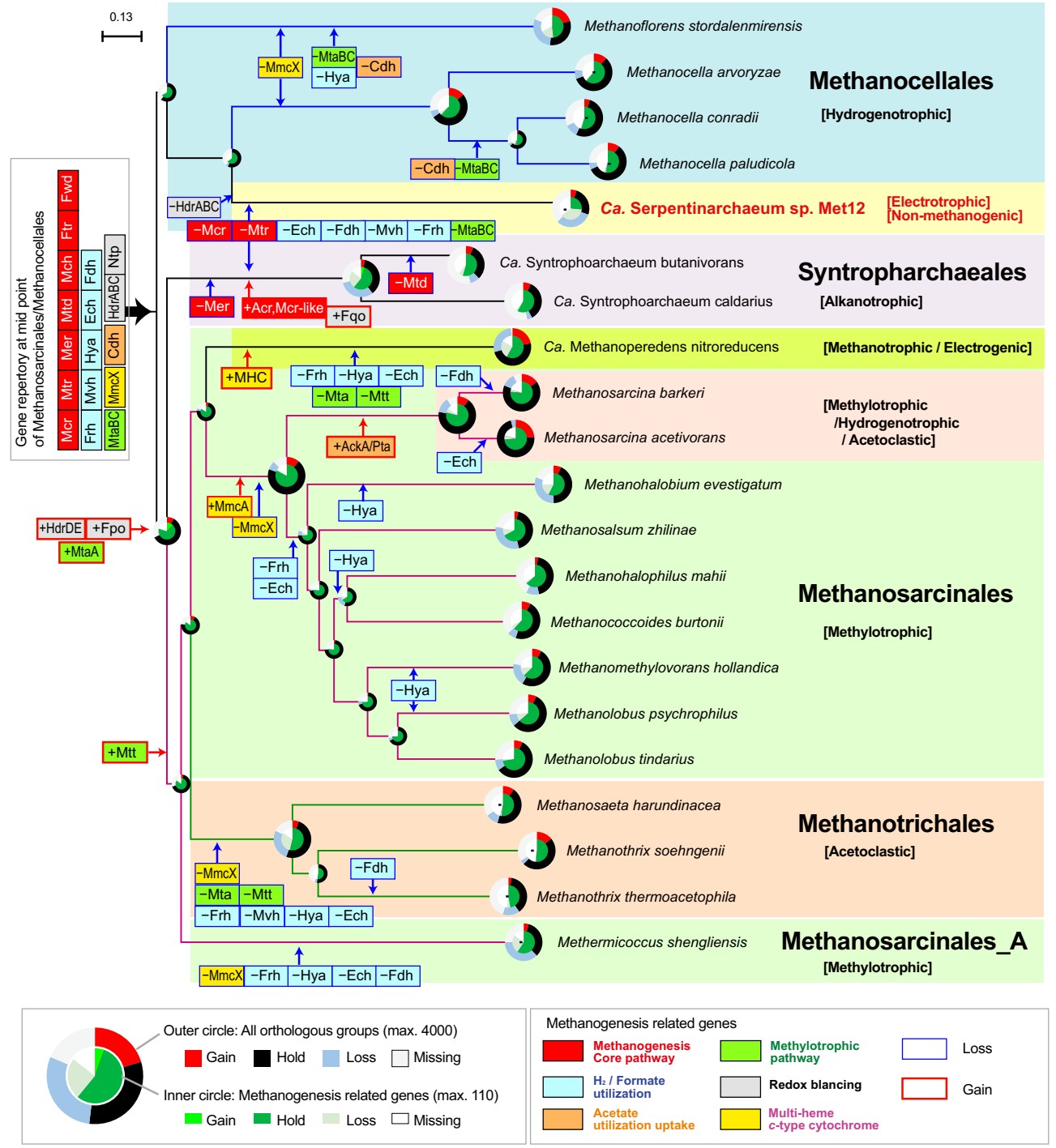

**Fig. 8 | Gain and loss of orthologous genes and methanogenesis-related genes.** Numbers of orthologous genes for gain, loss, hold, and missing were shown by pie charts at the branch point of unrooted phylogenomic tree of orders Methanocellales, Methanosarcinales, Syntropharchaeales, and Methanotrichales genomes. From the 110 methanogenesis-related genes (inner pie chart), specific key gene sets of the methanogenic process described in Fig. 1 were depicted in the colored boxes with + as gained gene set (red line) or − as lost gene set (blue line). Multi-heme c-type cytochromes that is not assigned as the key methanogenesis genes were also shown in yellow boxes.

101 bp PE for GPS1 and BS5 sample and as the 151 bp PE for BS5sc sample by Illumina's standard protocol. The RNA sequences have been deposited in the NCBI SRA under accession numbers SRX17443946 for GPS1 in 2011, SRX17443944 for BS5 in 2011, SRX17443945 for BS5 in 2012, and SRX17428102 for BS5sc in 2014.

Metagenomic reads from biofilm of hydrothermal field in Prony Bay Hydrothermal Field (SRX748870 and SRX748869 for chimney ST09), in Votri Massif travertine (SRX1055082 for spring BR2), in Santa Elena

Ophiolite alkaline spring (SRX748871 for Spring9), in Cabeco de Velle (SRX748868 for spring AC3), and in Old City hydrothermal vent (SRX6579314 for chimney BaC and SRX6579317 for chimney PiMo) were used for de novo assembly of CLC Genomic Workbench v8.6 (QIAGEN, Venlo, Netherlands) with default parameters. MAG sequence of *Methanocellales* archaeon MAG-838 (JAJOKI000000000.1) recovered from the metagenome of venting fluids in Lost City hydrothermal field was also used for the comparative analysis. Assembled scaffolds recovered from

metagenomes of serpentinized groundwaters pumped from three wells (BA1A, NSHQ14, WAB188) in The Samail Ophiolite, Sultanate of Oman (JGI IMG database under accessions 3300045453, 3300045950, 3300045482, and 3300045452) were also used for the comparative analysis. The global map provided from NOAA National Centers for Environmental Information as ETOPO 2022 15 Arc-Second Global Relief Model (https://doi.org/10.25921/fd45-gt74. Accessed 2024/03/08) was used to indicate the location of these sampling sites.

## Genome refinement and circulation of Met12

A MAG of Methanosarcinales archaeon Met12, recovered from the BS5-2012 metagenome[11] was used as a template for the further genome refinement in this study. The scaffolds of the MAGs were cleaved to contigs at the gap regions. Mate-pair reads (average distance between reads was ~2000 bp) were mapped to the contigs with the settings as 0.5 of minimum length and 0.95 of minimum similarity fractions using Map Reads to Reference algorism in CLC Genomics Workbench (version 8.6). The potential connections of contigs were analyzed by Collect Paired Read Statistics tool in CLC Genome Finishing Module (QIAGEN, Venlo, Netherlands). Based on the potential connections, the contigs were manually connected by using Align Contigs tool after the extension of contig edge by using Extend Contig tool in CLC Genome Finishing Module. After the manual curation, in order to polish contigs, metagenomic reads of BS5-2011, BS5-2012, and BS5sc-2014 were mapped to the contigs with the settings as 0.7 of minimum length and 0.95 of minimum similarity fractions, and the consensus sequences were extracted. Through the refinement of MAGs, the genome of Met12 was upgraded to only one circulated contig from 47 scaffolds. The taxonomic classification of the Met12 was assigned by both the concatenated alignment of single-copied housekeeping genes at GTDB-tk platform[55] and 16S rRNA at ARB-SILVA database[56]. The refined MAGs were deposited in NCBI under Biosample SAMN05965738 (accession number, CP017966).

A BLAST Ring image generator (BRIG)[57] was employed for visualizing a genome as a circular image and for comparison of the MAG Met12 from The Cedars spring with the metagenomic contigs of ST09 from Prony Bay hydrothermal field[13], Old City hydrothermal field[15], and MAG-838 from Lost City hydrothermal field[12].

## Functional annotation

Open reading frame (ORF) calling and functional annotation for MAG of Met12 were performed in NCBI prokaryotic genome annotation pipeline[58]. KEGG Automatic Annotation Server (KAAS) was used for the KEGG orthologous (KO) group assignment with the SBH (single-directional best hit) method set to 37 as the threshold assignment score[59]. ORFs were assigned to the archaeal Clusters of Orthologous Groups of proteins (arCOGs) by the best BLAST hit to the reference data[60] using an e-value cutoff of $1e^{-6}$. Localization of the proteins was analyzed by prediction of transmembrane helixes in TMHMM server version 2.0[61]. Taxonomic assignment of each ORF was analyzed by using GhostKOALA[62]. ORFs encoding c-type cytochromes were identified based on a CXXCH motif search as for covalent heme-binding domain[32], and ORFs contained more than two occurrences of the motif indicating multi-heme c-type cytochromes (MHcytCs). All the functional annotations performed in this study were summarized in the Supplementary Data 5.

## mRNA read mapping to ORFs

RPKM (Reads Per Kilobase per Million mapped reads) values[63] for both DNA and mRNA samples were separately determined by the RNA-Seq Analysis function in CLC Genomics Workbench (version 8.6). The nucleotide sequences of Met12 ORFs were used as references, and read mapping was performed using 0.5 as the minimum length and 0.95 as the minimum similarity fractions. In situ gene expression level was calculated by mRNA-RPKM per DNA-RPKM for each coding sequences (CDSs). The rank of the gene expression levels of Met12 was calculated by the average of mRNA-RPKM / DNA-RPKM among four mRNAseqs (BS5-2011, BS5-2012, BSsc-2014, and GPS1-2011). As for the BLAST Ring image generator (BRIG), the SAM files of the mRNA read mapping for BS5-2011, BSsc-2014, and GPS1-2011 were applied to generate coverage graph.

## Ortholog analysis

As for in house orthologous clustering among a MAG Met12, order Methanocellales (four species), and order Methanosarcinales (sixteen species) (Supplementary Data 2), OrthoMCL software was used to cluster the CDSs based on sequence similarity via an all-against-all BLAST search[64] on KBase platform[65]. From 53,750 CDSs assigned at RefSeq genome for 19 methanogenic/methanotrophic species, at IMG only for *Methanoflorens stordalenmirensis*, and at PGAP for MAG Met12, a total of 44,963 proteins (83.7% of the total dataset) were clustered into 13,715 ortholog groups (MS orthologs), while 8,787 CDSs were singleton (Supplementary Data 4). Functional annotation for the 53,750 CDSs was performed on similar way to the CDSs of MAG Met12. Average values of the heme-binding domains and the transmembrane helixes were calculated for each MS ortholog. The core metabolic marker genes for methanogens were chosen based on ko00680 (methane metabolism) at KEGG pathway database. Several transporters, translocases, and oxidoreductases related to the methanogenic/methanotrophic reactions (Fpo, Fqo, Rnf, Mrp) are also picked as key metabolic marker genes. The phylogenomic tree with MAG Met12 was constructed and exported by using SpeciesTreeBuilder on KBase platform. From the matrix of the ortholog table, a Dollo parsimony model[66] was applied with the default parameters of the COUNT package[67] for inferring when genes were gained or lost. The number of orthologs gained or lost was visualized on a pie chart on each branch of the phylogenomic tree using iTOL version 6[68]. All the MS orthologs were summarized in Supplementary Data 3.

## Phylogenetic tree analyses and 3D structure prediction

Amino acid sequence of MmcX (ATZ61495.1) in the MAG Met12 was blasted against nr database. Forty-seven closest amino acid sequences were retrieved from the database for applying the tree construction. The 3D structure prediction using AI was performed for MmcX in the Met12 by using ColabFold[69] at the default settings for running AlphaFold2[70]. The PDB file was loaded into PyMOL [https://pymol.org/2/] to visualize the structure. Signal peptides and the cleavage site was predicted using SignalP v6[71], and the signal peptide of the MmcX was disappeared in the protein structure. The protein structure similarity between AvECN (AEA46122) and MmcX in the Met12 was calculated by the RMSD after alignment of the protein structures in PyMOL.

Amino acid sequences of SbtA (K07086) in the Met12 (ATZ60936.1) and in the metagenomes from the other serpentinization fields were extracted, while reference sequences described in the previous manuscript[72] were also retrieved from the database. MUSCLE[73] was used for the sequence alignment, and Maximum Likelihood with RaxML[74] was used for the tree construction.

## Bacterial strains, plasmids, and growth conditions

*Shewanella oneidensis* strains were cultivated at 30 °C in lysogeny broth (LB) or lactate minimal medium (LMM) containing 15 mM lactate as the sole carbon and energy source[75]. For aerobic cultivation, 5 mL of LB medium or LMM in a test tube (30 mL capacity) was inoculated with an *S. oneidensis* strain at an initial optical density at 600 nm ($OD_{600}$) of 0.05 and shaken at 180 rpm. For anaerobic cultivation, 8 mL of LMM supplemented with 30 mM fumarate in a screw-top test tube (9 mL capacity) was inoculated with an *S. oneidensis* strain at an initial $OD_{600}$

of 0.01. Test tubes containing the anaerobic cultures were capped with Teflon-coated butyl rubber septa, sealed with aluminum crimp seals, and purged with high-purity nitrogen gas (99.99%). The growth of cells was monitored by measuring the $OD_{600}$ of cultures using a UH5300 spectrometer (Hitachi, Tokyo, Japan) or a mini photo 518 R (Taitec, Tokyo, Japan). *Escherichia coli* strains were cultivated in LB medium at 37 °C. When necessary, 50 μg/mL kanamycin (Km) was added to culture media. Agar plates contained 1.6% Bacto Agar (Difco, Franklin Lakes, NJ, USA).

## Construction of mutants
To construct pHSG-mmcX, the DNA fragment containing the *mmcX* gene whose codon usage was optimized for expression in *S. oneidensis* (Supplementary Fig. 4a) was chemically synthesized and cloned between the BamHI and EcoRI sites in pHSG298 (Takara, Tokyo, Japan) (Supplementary Fig. 4b). The resultant plasmid pHSG-mmcX and the control vector pHSG298 were introduced into wild-type *S. oneidensis* strain MR-1 and its derivatives (ΔomcAΔmtrC and ΔmtrA)[38] by electroporation according to a method described elsewhere[76].

## Operation of electrochemical cells
A single-chambered three-electrode electrochemical cell (EC; 18 ml total capacity)[77] was used to monitor electric current generated by *S. oneidensis* derivatives under potential-controlled conditions. The EC was equipped with a graphite felt working electrode (2.25 cm$^2$), Ag/AgCl reference electrode (+0.199 V vs. the standard hydrogen electrode; SHE) (HX-R5, Hokuto Denko, Tokyo, Japan), and platinum wire counter electrode (10 cm, φ0.3 mm; Nilaco, Tokyo, Japan). The EC was filled with 15 mL of an electrolyte solution containing 9 mM $(NH_4)_2SO_4$, 5.7 mM $K_2HPO_4$, 3.3 mM $KH_2PO_4$, 100 mM NaCl, and 30 mM HEPES-NaOH buffer (pH 7.4), and inoculated with bacterial cells at an initial $OD_{600}$ of 0.1. Current was monitored using a multichannel potentiostat (VMP3; Biologic, Claix, France), and current density (μA/cm$^2$) was calculated based on the projected area of the working electrode. To evaluate the ability of *S. oneidensis* derivatives to receive electrons from low-potential electrodes, the working electrode was poised at −400 mV vs. SHE. After the EC was incubated until current became stable (for approximately 2 h), 20 mM fumarate was added to the electrolyte as an electron acceptor to measure cathodic current. To evaluate the ability of *S. oneidensis* derivatives to transfer electrons to high-potential electrodes, the working electrode was poised at +200 mV vs. SHE, and anodic current was measured using the electrolyte supplemented with 15 mM lactate as an electron donor.

## Reporting summary
Further information on research design is available in the Nature Portfolio Reporting Summary linked to this article.

## Data availability
All unassembled sequences related to this study have been deposited in the NCBI Sequence Read Archive under accession numbers SRX17443947, SRX17443946, SRX17443944, SRX17443945, and SRX17428102 on BioProject PRJNA351917. The circulated genome data of MAG Met12 has been submitted in the NCBI GenBank under accession number CP017966. The detailed data generated in this study are provided in the Supplementary Data or Source Data files. Raw analytical data of Met12 CDSs generated in this study are provided in the Supplementary Data 5, while the source data of the orthologous analyses for the comparative genomes are provided in the Supplementary Data 3. Source data are provided with this paper.

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

## Acknowledgements
We greatly appreciate Mr. Roger Raiche and Mr. David McCrory for providing their private land for our research. We appreciate Dr. Tomomi Sumida for the technical advice of 3D structure prediction and visualization, Prof. J. Gijs Kuenen, Dr. Ken Takai, Prof. Keishi Okazaki and Prof. Tomoyuki Kosaka for the constructive discussions. This work was supported by JST, CREST Grant Number JPMJCR20S4 and JSPS KAKENHI Grant Numbers 22H05152, 22H00429, 20H03314, and 20H04620.

## Author contributions
S.S. and S.I. conceptualized and designed the study. S.S. and S.I. participated in the sampling campaign and S.S. recovered and processed the original samples. S.S. performed DNA and RNA extraction. S.S., S.I., G.L.C., M.A. and P.H.N performed metagenomic and metatranscriptomic analyses. Y.T., A.K. and K.W. performed electrochemical analysis. S.S., S.I., G.L.C., F.I. and K.H.N. wrote the manuscript with contributions from all coauthors.

## Competing interests
The authors declare no competing interests.
