## [Peer Review File · Nature Communications]

A non-methanogenic archaeon within the order MethanocellalesEditorial Note: Parts of this Peer Review File have been redacted as indicated to remove third-party material where no permission to publish could be obtained.

Reviewer #1 (Remarks to the Author):

Until 2015, all known methanogens belong to the phylum Euryarchaeota. This view has been drastically changed since to include other archaeal phyla, but much of the evidence relies on finding genes such as methyl-coenzyme M reductase (*mcr*), which are so essential for the organismal lifestyle that these genes became the defining characteristic for their associated microbial groups. *Candidatus Serpentinarchaeum* reported here by Suzuki and colleagues becomes the first reported exception to this widely accepted view: while this novel genus belongs to the traditional methanogenic order Methanocellales, it does not contain any of the essential methanogenesis genes. The authors circularized the genome of a representative in this genus, Met12, obtained from a serpentinization site The Cedars in Northern California to prove their point. Moreover, the authors explored the possible lifestyle of this novel microorganism using metatranscriptomics and heterologous expression. The underlying data presented in the study is excellent and the finding is of general interest to the readers of Nature Communications. Below are some questions and suggestions for the authors to consider and may further strengthen their study.

Questions/suggestions:

- 1) The authors hypothesize that Met12 is an acetogen as opposed to an acetotroph primarily based on their heterologous expression result. While the heterologous expression and electrochemical experiment suggest this is the case, the results may not rule out that Met12 could be an acetotroph. Could the MmcX of Met12 be bidirectional, and the expression of an archaeal protein in a bacterial host altered its function such that it no longer could be used for electron export? This becomes even more puzzling since Met12 uses an unconventional route for the conversion of acetyl-CoA and acetate: instead of using Ack/Pta as most acetogens, Met12 uses Acs (L217). What is the physiological and energetic implication using Acs instead of Ack/Pta for acetogenesis (or acetotrophy)?
- 2) Multiheme cytochrome c could be used for syntrophy, as the authors briefly noted in L155, but did not mention this possibility in the rest of the manuscript such as L144, L281 and Figure 4c. Could Met12 be a syntroph?
- 3) The authors mention in supplementary text that the MmcX's redox potential could also be used to indicate its function. To this point, has the authors measured the cyclic voltammetry of Met12-MmcX expressing *Shewanella*?
- 4) In supplementary information, the authors state that "MHcytCs with S-layer domains commonly found in Euryarchaeota were absent in Met12 and *Ca. Methanoflorens stordalmirensis*, indicating that, in contrast to the situation with ANME, the EET reaction does not require the large MHcytC that penetrates the S-layer". *Methanosarcina* with MmcA presumably has EET capability but does not have MHcytC that penetrate the S-layer (see Figure 1A in this manuscript), so I do not think the logic here is valid.
- 5) If MmcX has a membrane-binding site (L139), how could it form a nanowire (L151)?
- 6) What is the acetate concentration in The Cedars or other serpentinization environments?
- 7) Figure 6: could this tree be rooted and what is the ancestral gene set? Instead of a circular tree, maybe it is clearer to show this tree in the rectangular format as Figure 1?

Minor issues:

- 1) Please go over the manuscript carefully to fix typos (e.g. L62: "polyextreme conitions", L249: "ca. Serpentinarchaeum" should italicize and capitalize "Ca.") and italicize organism and gene names (e.g. Figures and Tables).
- 2) Please clarify "median" mentioned in L138 and L198 refer to all transcripts or just those of Met12?
- 3) The authors claim that Met12-like genomes were detected in 6 ocean metagenomic datasets on L227. However, in Fig. 5, there are 7 serpentinization sites shown with Met12-like genomes absent in 3 sites. How many metagenomes did the authors investigate? Additionally, how could 1 genome, MAG-838, be used to show "the absence of Mcr, associated methanogenesis proteins and Hdr, appearing to be a common feature of these serpentinization inhabitants" in L231? What about other MAGs?
- 4) L85: define dDDH
- 5) L95: "the reaction". What reaction?
- 6) L166+167: gene name vs protein name

- 7) L190: define DIC
- 8) L208: "effectuate"?
- 9) L237L "Halobacteriota" is mentioned here and elsewhere, as well as "Euryarchaeota". Please be consistent with taxonomy naming.
- 10) Figure 1A: the phylogenetic tree is based on what gene?
- 11) Figure 3a: what kind of phylogenetic tree?
- 12) L353: Ca. Methanopredens nitroreducens. Spelling and italicization incorrect.
- 13) L400: should be MHcytC to be consistent
- 14) L417: phylogenetic tree of proteins instead of genes.
- 15) L592: -0.6V vs SHE mentioned here, while the main text L400 mentioned -400 mV vs SHE.

Reviewer #2 (Remarks to the Author):

The study "Electron fueled acetogenic archaea in highly reduced serpentized setting provides insight into the primordial carbon fixation" discovered a methanogenic archaeal member capable of utilizing the acetyl-CoA pathway but lacking essential genes for methanogenesis.

General comments

I enjoyed reading the manuscript. It will be more than suited for the journal, and I recommend accepting the manuscript. I do not have much to add. The manuscript presents significant data for a broad scientific audience. The discovery is novel and essential for many scientific and industrial platforms. The methods are sound. The manuscript is clearly outlined and well-written, and the reader is taken along the journey by easily understandable, delightful Figures.

Reviewer #3 (Remarks to the Author):

Suzuki report several MAGs associated with a Methanosarcinales affiliated putative methanogen in the Cedars over several years and across several sites. Similar MAGs were later identified in other marine serpentizing systems. Interestingly, the MAGs lacked genes for key elements of traditional methanogenesis pathways such as a variety of [NiFe]-hydrogenases, Mtr, Mcr, and heterodisulfide reductase. This suggested an alternative energy (and carbon) metabolism in these cells. Evidence is presented to indicate it is possibly an acetogen (has the requisite proteins). More interestingly, the MAGs encode polyheme cytochromes that are putatively extracellularly oriented (signal peptide) that are electrogenic. Heterologous expression indicates these cytochromes are most likely involved in acquisition of electrons.

Overall, I found many of the observations to be of high interest to the broad community interested in serpentizing environments. The informatics, phylogenetics, and biochemical experiments appear sound. However, there are elements of their interpretation that I find to be a bit oversold, including the proposed relevance to the origin of carbon fixation in serpentizing systems. Further, I have several questions regarding the interpretation of the metabolic model that is presented. The comments below are meant to further improve this paper:

Lines 2 and 3: There is really not much primordial about these organisms. Early evolving methanogens and acetogens (I believe) do not encode cytochromes and the current study did not really address the overarching question regarding primordial C fixation which is how it might be obtained in such an environment. The electroactive cytochromes do not necessarily help with this. My preference would be to focus the study on what it is – a very interesting set of adaptations to more efficiently capture reducing equivalents to drive acetogenic metabolism in a methanogen.

Lines 47 and 48: Cannot produce electrons as they are not free nor can they be created or destroyed

Lines 51 and 52: Formate is an intermediate during the reduction of CO₂ to methane. Sentence needs to be restructured.

Lines 52 and 53: Need to introduce the WL pathway prior to this and its semblance to the serpentinization reaction since there is no connection between these statements and acetogens up to this point (a topic of the next sentence)

Line 55: Starting

Lines 71-78: Where is Met12 in all of this? In lines 82-83, it is stated that it is the dominant archaeon but from the introductory text in these lines, I don't get the same impression

Lines 88-91: The small genome is consistent with genome streamlining, which has been seen in the Cedars and Oman before, and when combined with the phylogeny, is consistent with gene loss being a derived trait. Thus, the relevance of this organism or metabolic strategy to primitive life is questionable.

Lines 92-121: How complete are the MAGs? What are the levels of their contamination? How well were unbinned contigs screened for potential homologs of these key enzymes?

Line 97: Which order does Met12 belong to – not definitively stated.

Line 110: Technically, and per Fig. 1, it produces formylmethanofuran not CO. CO (or Cdh) is not involved in the first step of methanogenesis.

Line 113: Fd is also reduced by coupled ion transport as well (Eha/Ehb/Ech).

Line 115: Again, this indicates it is a recent adaptation. This is interesting but does make it of little relevance to early Earth studies.

Line 125-126: What would be the utility of the acetate transporter? To exclude it from the cell? That is something that should be played up in this as a potential source of organic carbon to fuel the rest of the community (per line 221)

Line 118-119: What is the evidence that it does not interact with Fd or Hdr? Also, why is it depicted on the outside of the cell's membrane? Is there a signal peptide that would get it there? What would be the electron donor/acceptor in this case? What its function be? Most NiFe hydrogenases in Methanosarcinales (minus Vht) are cytoplasmic or at most inner membrane associated (see Kulkarni 2018 for a model).

Line 135: Do you mean pools? What conditions are being referred to here?

Lines 137-138: Why would one expect these to be so highly expressed? Are they expected to turnover often and thus need to be continually resynthesized?

Lines 185-186: is this also upregulated in these cells? It should be, if one is to consider the stoichiometry of the WL pathway electrons and energy (carbon) substrates

Lines 187-188: This is misleading. They pull the solubility of carbonate into solution by consuming the vanishingly small amount of bicarbonate in solution. Also, carbonates are highly restricted to the near surface portion of ophiolites where higher reduction potentials are commonly encountered. The same could be true for the Cedars. Does this change this interpretation?

Per the model in Fig. 4, it seems to suggest that RNF is running in reverse (to reduce Fd and NAD) which would drive ion transport into the cell. It should not be depicted as being reversible if this is truly how the authors think the polyheme cytochromes are operating (electron extraction). This is

stated by the authors as such in lines 207-209. This makes me confused as to how the ion gradient to power ATP synthesis via the encoded ATPase is formed.

Lines 81-252: There is a lot of speculation and interpretation in the results section that might be easier to synthesize by moving to the discussion.

Line 249-250: You have not prepared the reader yet to understand why loss of bifurcating capabilities might be an adaptation to hyper reduced environments

Lines 251-252: These are not found in deeply rooted archaeal methanogens; not sure about the deeply rooted acetogens.

Line 264: Which lineage, Archaea? There have been several reports of putative archaeal acetogens from hypersaline environments recently.

Line 280: While I appreciate the heterologous biochemical work conducted herein, I don't think it demonstrates that this is what is happening in Met12 to the level of confidence indicated in this sentence.

Lines 281-282: The use of CO₂ was not demonstrated either, nor was EET. Just in the heterologous expression work.

Line 294: highly reduced minerals

Line 287-298: I don't think this work has relevance to what primitive methanogens were doing given the aforementioned comments about this being a clear example of gene loss leading to this putative phenotype. Further, deeply rooted methanogens do not encode cytochromes, making it a stretch to link these observations to early evolving members of this lineage.

REPLY TO THE REVIEWER COMMENTS

Reviewer #1:

Until 2015, all known methanogens belong to the phylum Euryarchaeota. This view has been drastically changed since to include other archaeal phyla, but much of the evidence relies on finding genes such as methyl-coenzyme M reductase (*mcr*), which are so essential for the organismal lifestyle that these genes became the defining characteristic for their associated microbial groups. *Candidatus Serpentinarchaeum* reported here by Suzuki and colleagues becomes the first reported exception to this widely accepted view: while this novel genus belongs to the traditional methanogenic order Methanocellales, it does not contain any of the essential methanogenesis genes. The authors circularized the genome of a representative in this genus, Met12, obtained from a serpentinization site The Cedars in Northern California to prove their point. Moreover, the authors explored the possible lifestyle of this novel microorganism using metatranscriptomics and heterologous expression. The underlying data presented in the study is excellent and the finding is of general interest to the readers of Nature Communications. Below are some questions and suggestions for the authors to consider and may further strengthen their study.

We appreciate your comment.

Questions/suggestions:

1) The authors hypothesize that Met12 is an acetogen as opposed to an acetotroph primarily based on their heterologous expression result. While the heterologous expression and electrochemical experiment suggest this is the case, the results may not rule out that Met12 could be an acetotroph. Could the MmcX of Met12 be bidirectional, and the expression of an archaeal protein in a bacterial host altered its function such that it no longer could be used for electron export? This becomes even more puzzling since Met12 uses an unconventional route for the conversion of acetyl-CoA and acetate: instead of using Ack/Pta as most acetogens, Met12 uses Acs (L217). What is the physiological and energetic implication using Acs instead of Ack/Pta for acetogenesis (or acetotrophy)?

While the possibility of Met12 being an acetate-oxidizing and iron-reducing archaeon cannot be entirely ruled out, it seems unlikely considering the results from the heterologous expression of MmcX and electrochemical experiments, as well as the excessively reducing and alkaline environmental conditions which are unsuitable for microbial solid iron (mineral) reduction. Certain homoacetogens can reverse the acetyl-CoA pathway and become acetotrophs, but this process requires a symbiotic partner capable of consuming hydrogen. For instance, *Thermacetogenium phaeum* strain PB is a homoacetogenic bacterium, yet it only oxidizes acetate when grown with a symbiotic partner or when a hydrogen concentration is below 0.22 μM . (Ishii et al. 2006, AEM DOI: <https://doi.org/10.1128/AEM.00333-06>). Therefore, acetate oxidation through the reversal of the acetyl-CoA pathway appears unlikely due to the water in The Cedars being saturated with hydrogen and rich in electrons.

Related to the Ack/Pta or Acs for the final step of acetogenesis, we cited two papers, (Ouboter et al. and Schlegel et al.) which showed the reversible reaction of Acs. In addition, a recent publication of PNAS (Colman et al. 2022) reported the potential acetogens in *Ca. Acetothermia* from an active serpentinizing site in Oman. In the publication, they described that “ Given the minimal energy conserved during acetogenesis, phosphorylation of acetyl-CoA produced from the WL pathway by phosphotransacetylase (Pta) and its subsequent dephosphorylation to produce one ATP and acetate by acetate kinase (AckA) has been traditionally considered necessary for acetogens to balance the consumption of one ATP molecule from HCOO^- fixation in the methyl branch of the WL pathway, thereby rendering the WL pathway ATP consumption neutral. However, as observed elsewhere for other *Acetothermia*, genes encoding AckA or Pta were not identified in the genomes of either type I or type II *Acetothermia*. Rather, a homolog of an archaeal-like ADP-forming acetyl-CoA synthetase (ACD) was identified in the type I MAGs but not in the type II MAGs. Further, both the type I and type II MAGs encoded homologs of the non-ATP generating acetyl-CoA synthase [Acs, also known as the AMP-forming Acs;

abbreviated here as ACS to differentiate from the Acs discussed above] that is generally considered an assimilatory, but potentially reversible, acetate utilization mechanism.” We think that the same situation occurs in the Acs of Met12. We added this discussion briefly in the revised manuscript (Line 289-306).

2) Multiheme cytochrome c could be used for syntrophy, as the authors briefly noted in L155, but did not mention this possibility in the rest of the manuscript such as L144, L281 and Figure 4c. Could Met12 be a syntroph?

The Met12 is unlikely a syntroph of other bacteria because potential syntrophic bacterial partners, i.e. sulfate reducers, nitrate reducers, iron reducers, were not detected in the deep groundwater of The Cedars serpentized site as we reported previously (Suzuki et al. ISME J 11, 2584-2598 (2017)). We searched genes responsible for the bacterial respiration (e.g., electron transport chain and terminal electron acceptor reductases) in the entire metagenomic data for deep groundwater but we could not detect those. Therefore, there is no potential symbiotic partner of Met12 in The Cedars deep groundwater ecosystem (Line 297-299).

3) The authors mention in supplementary text that the MmcX’s redox potential could also be used to indicate its function. To this point, has the authors measured the cyclic voltammetry of Met12-MmcX expressing *Shewanella*?

We have been attempting to determine the redox potential of MmcX through cyclic voltammetry, but it has proven to be challenging thus far. The specific range we aim to explore is within an alkaline and highly reducing environment. This pursuit presents two main challenges: the need to mitigate the effects of chemical hydrogen production and the low alkaline tolerance of the host *Shewanella*, for which reliable data have not yet been obtained. We continue our efforts to overcome these challenges.

4) In supplementary information, the authors state that “MHcytCs with S-layer domains commonly found in Euryarchaeota were absent in Met12 and *Ca. Methanoflorens stordalmirensis*, indicating that, in contrast to the situation with ANME, the EET reaction does not require the large MHcytC that penetrates the S-layer”. *Methanosarcina* with MmcA presumably has EET capability but does not have MHcytC that penetrate the S-layer (see Figure 1A in this manuscript), so I do not think the logic here is valid.

We agree with the reviewer’s comments. We removed the discussion about the MHcytC that penetrate the S-layer.

5) If MmcX has a membrane-binding site (L139), how could it form a nanowire (L151)?

The structure of Archaeal nanowires, recently published in *Cell* (186, 13, 2853 - 2864.e8 (2023)), revealed a significant similarity in amino acid sequence between MmcX and the avECN reported in that study. We conducted a comparison of the predicted structures of avECN and MmcX, where the avECN is represented in red and the MmcX-Met12 in green. The lightly shaded regions denote the signal peptide areas, which undergo cleavage as they traverse the membrane. Notably, the structures of the N-terminal transmembrane region and the cleavage site are identical, suggesting a similar localization for both proteins. This implies that MmcX is also presumably a nanowire protein. We added this information to the Discussions of in situ gene expression analysis: 1) Multi-heme c-type cytochrome, MmcX in the supplementary discussion.

6) What is the acetate concentration in The Cedars or other serpentinization environments?

Acetate concentration is about 70 μM in The Cedars GPS1 (*ISME J* 17, 95–104 (2023)). We added the acetate concentration in the revised manuscript (Line 73).

7) Figure 6: could this tree be rooted and what is the ancestral gene set? Instead of a circular tree, maybe it is clearer to show this tree in the rectangular format as Figure 1?

This tree is unrooted. We changed the phylogenetic tree in the rectangular format.

Minor issues:

1) Please go over the manuscript carefully to fix typos (e.g. L62: “polyextreme conitions”, L249: “ca. Serpentiarchaeum” should italicize and capitalize “Ca.”) and italicize organism and gene names (e.g. Figures and Tables).

Thank you for pointing out the typos. We carefully fixed typos.

2) Please clarify “median” mentioned in L138 and L198 refer to all transcripts or just those of Met12?

This shows just the transcripts of Met12, and the median means the median of Met12’s transcripts. We added the explanation (Line 151).

3) The authors claim that Met12-like genomes were detected in 6 ocean metagenomic datasets on L227. However, in Fig. 5, there are 7 serpentinization sites shown with Met12-like genomes absent in 3 sites. How many metagenomes did the authors investigate? Additionally, how could 1 genome, MAG-838, be used to show “the absence of Mcr, associated methanogenesis proteins and Hdr, appearing to be a common feature of these serpentinization inhabitants” in L231? What about other MAGs?

We initially analyzed seven sites, including The Cedars. However, we included one additional site during this revision. Consequently, a total of eight sites were examined in this study: three oceanic sites (Lost City, Old City, New Caledonia) and four terrestrial sites (Cabeço de Vide, Voltri Massif, Santa Elena Ophiolite, as well as Samail Ophiolite), in addition to The Cedars (Line 238-241).

Discussing the absence of genes from a genome can be particularly challenging because they might be present in contigs that were not utilized for the MAGs. To address this issue, we circularized the genome of Met12 and further confirmed the absence of those genes in contigs that were not used for MAGs (See Validation of lack of Mcr, Mtr, and Hdr in the MAG Met12 in Supporting Information). While the sequence of MAG-838 from the Lost City has been deposited in GenBank and published (Brazelton, *Appl Environ Microbiol*, e0092922 (2022)) and mentioned the absence of Mcr and Hdr in the Supplemental Material, the remaining MAGs were constructed by our team. However, it's important to note that the quality of these MAGs may not always be sufficiently high to allow for discussions at the same level of detail as with Met12. Additionally, we avoided thoroughly analyzing the metagenomic data of serpentinized ecosystems due to a potential risk of interfering with the work of other scientific teams.

4) L85: define dDDH

Corrected as suggested.

5) L95: “the reaction”. What reaction?

Changed the reaction to the enzyme.

6) L166+167: gene name vs protein name

Corrected as suggested.

7) L190: define DIC

DIC was defined above (Line 64)

8) L208: “effectuate”?

I modified the sentence to provide clarification.

9) L237L “Halobacteriota” is mentioned here and elsewhere, as well as “Euryarchaeota”. Please be consistent with taxonomy naming.

There are two references to Euryarchaeota in the text, but both are provided with their previous names in parentheses. While we aimed to maintain consistency in taxonomy naming for our readers, some software we utilized (e.g., GhostKOALA) has not updated the taxonomic names during this transitional phase. Consequently, we were unable to do so ourselves.

10) Figure 1A: the phylogenetic tree is based on what gene?

This is a concatenated phylogenomic tree based on marker gene sets. The method is described in the ortholog analysis section in

11) Figure 3a: what kind of phylogenetic tree?

MmcX and its homologous proteins. We added the explanation in the manuscript.

12) L353: Ca. Methanopredens nitroreducens. Spelling and italicization incorrect.

Corrected as suggested.

13) L400: should be MHcytC to be consistent

Corrected as suggested (Moved to Supporting information).

14) L417: phylogenetic tree of proteins instead of genes.

Corrected as suggested.

15) L592: -0.6V vs SHE mentioned here, while the main text L400 mentioned -400 mV vs SHE.

We apologize for the confusion. The value of - 400 mV vs SHE is correct. This confusion was caused in the manuscript because -0.6 vs Ag/AgCl₂ was frequently used in our lab.

Reviewer #2:

The study “Electron fueled acetogenic archaea in highly reduced serpentinized setting provides insight into the primordial carbon fixation” discovered a methanogenic archaeal member capable of utilizing the acetyl-CoA pathway but lacking essential genes for methanogenesis.

General comments

I enjoyed reading the manuscript. It will be more than suited for the journal, and I recommend accepting the manuscript. I do not have much to add. The manuscript presents significant data for a broad scientific audience. The discovery is novel and essential for many scientific and industrial platforms. The methods are sound. The manuscript is clearly outlined and well-written, and the reader is taken along the journey by easily understandable, delightful Figures.

We appreciate your comment. We are glad you enjoyed it; it was very encouraging for us.

Reviewer #3:

Suzuki report several MAGs associated with a Methanosarcinales affiliated putative methanogen in the Cedars over several years and across several sites. Similar MAGs were later identified in other marine serpentinizing systems. Interestingly, the MAGs lacked genes for key elements of traditional methanogenesis pathways such as a variety of [NiFe]-hydrogenases, Mtr, Mcr, and heterodisulfide reductase. This suggested an alternative energy (and carbon) metabolism in these cells. Evidence is presented to indicate it is possibly an acetogen (has the requisite proteins). More interestingly, the MAGs encode polyheme cytochromes that are putatively extracellularly oriented (signal peptide) that are electrogenic. Heterologous expression indicates these cytochromes are most likely involved in acquisition of electrons.

Overall, I found many of the observations to be of high interest to the broad community interested in serpentinizing environments. The informatics, phylogenetics, and biochemical experiments appear sound. However, there are elements of their interpretation that I find to be a bit oversold, including the proposed relevance to the origin of carbon fixation in serpentinizing systems. Further, I have several questions regarding the interpretation of the metabolic model that is presented. The comments below are meant to further improve this paper.

We appreciate your comments.

Lines 2 and 3: There is really not much primordial about these organisms. Early evolving methanogens and acetogens (I believe) do not encode cytochromes and the current study did not really address the overarching question regarding primordial C fixation which is how it might be obtained in such an environment. The electroactive cytochromes do not necessarily help with this. My preference would be to focus the study on what it is – a very interesting set of adaptations to more efficiently capture reducing equivalents to drive acetogenic metabolism in a methanogen.

The hypothesis that serpentinized environments are analogous to early Earth has been widely reported (e.g., Schwander et al. (2023) *Front. Microbiol.* 14:1257597. doi: 10.3389/fmicb.2023.1257597). This is supported not only by geochemical studies, but also by chemical evolutionary studies. Based on this context, I initiated this research from a simple question: what drives and constrains the metabolism of life in early Earth analogue environments? It is a different approach from molecular phylogenetics, but we consider it as one approach to gradually connect the significant gap between chemical evolution and biological evolution, or geochemistry and

biochemistry. We think it not just simply a very interesting set of adaptations to capture more efficiently reducing equivalents to drive acetogenic metabolism in a methanogen, but at least, as one of the optimal solutions for carbon fixation in early Earth analogue environments.

However, as you pointed out, this study might be somewhat oversold because it lacks an evolutionary analysis from a chronological perspective. Therefore, some changes have been made to the title and the part of the abstract relating to primordial metabolism.

Lines 47 and 48: Cannot produce electrons as they are not free nor can they be created or destroyed

We removed electron.

Lines 51 and 52: Formate is an intermediate during the reduction of CO₂ to methane. Sentence needs to be restructured.

Formate is an intermediate in Fischer-Tropsch type reactions, but accumulation of formate can be observed in serpentinized systems (e.g. Formate concentration of The Cedars, Hakuba-Happo, Lost City are 7 μM, 8 μM and 36–158 μM respectively).

Lines 52 and 53: Need to introduce the WL pathway prior to this and its semblance to the serpentinization reaction since there is no connection between these statements and acetogens up to this point (a topic of the next sentence)

Corrected as suggested (Line 52-54).

Line 55: Starting

Corrected as suggested.

Lines 71-78: Where is Met12 in all of this? In lines 82-83, it is stated that it is the dominant archaeon but from the introductory text in these lines, I don't get the same impression

I changed the order of archaeal phylum (Line 81).

Lines 88-91: The small genome is consistent with genome streamlining, which has been seen in the Cedars and Oman before, and when combined with the phylogeny, is consistent with gene loss being a derived trait. Thus, the relevance of this organism or metabolic strategy to primitive life is questionable.

Although it is shown that Met12 has a small genome here, it is not intended to state the link between a small genome and primitive life.

Lines 92-121: How complete are the MAGs? What are the levels of their contamination? How well were unbinned contigs screened for potential homologs of these key enzymes?

MAG QC is shown in Extend Table 1 and searches for potential homologs of these key enzymes are described in the section of 'validation of lack of Mcr, Mtr and Hdr in the MAG Met12' in the supplementary discussion in the supporting information.

Line 97: Which order does Met12 belong to – not definitively stated.

Methanocellales. We stated the order (Line 94).

Line 110: Technically, and per Fig. 1, it produces formylmethanofuran not CO. CO (or Cdh) is not involved in the first step of methanogenesis.

Corrected the manuscript on Fwd metabolism, as noted by the reviewer (Line 115).

Line 113: Fd is also reduced by coupled ion transport as well (Eha/Ehb/Ech).

Corrected the manuscript as noted by the reviewer (Line 117).

Line 115: Again, this indicates it is a recent adaptation. This is interesting but does make it of little relevance to early Earth studies.

The absence of the complete FBEB (ferredoxin-based electron bifurcation) might represent a recent adaptation but specific to the early Earth analogue setting as well. One of the paradoxes in chemical evolution regarding the origin of carbon fixation via the acetyl-CoA pathway is how CO₂ fixation could have taken place before the existence of proteins enabling electron bifurcation. Thus far, carbon monoxide is recognized as a substrate enabling carbon fixation without FBEB. However, electron-driven CO₂ fixation could also be a viable alternative. In this context, we believe that these findings hold relevance for the study of early Earth.

Line 125-126: What would be the utility of the acetate transporter? To exclude it from the cell? That is something that should be played up in this as a potential source of organic carbon to fuel the rest of the community (per line 221)

Thank you for the comment. We added the discussion, and also included the concentration of acetate in The Cedars water (Line289-306).

Line 118-119: What is the evidence that it does not interact with Fd or Hdr? Also, why is it depicted on the outside of the cell's membrane? Is there a signal peptide that would get it there? What would be the electron donor/acceptor in this case? What its function be? Most NiFe hydrogenases in Methanosarcinales (minus Vht) are cytoplasmic or at most inner membrane associated (see Kulkarni 2018 for a model).

As shown in Supplementary Data1, orthologous analysis revealed that HyaABC and VhtAGC are orthologs. HyaAB or VhtAG is the active unit of the hydrogenase, while HyaC or VhtC transfers electrons to the MP on the membrane. However, Met12 does not have this HyaC (VhtC) therefore this hydrogenase cannot play a role in the reduction of Fdx in the cell. We added the explanation in the revised manuscript (Line 123-129).

Redacted

Line 135: Do you mean pools? What conditions are being referred to here?

Explanation of three sampling sites were added to the (L142-147). Photos in the Extended Data Fig. 1 would be easier to understand the conditions.

Lines 137-138: Why would one expect these to be so highly expressed? Are they expected to turnover often and thus need to be continually resynthesized?

This is an interesting question. Generally, groups of genes related to energy metabolism or to the translational system that are essential under certain conditions tend to be highly expressed. Rate-limiting factors in key

proteins of energy metabolism (e.g. Mcr and Dsr) tend to be particularly highly expressed. Pili and Flagellum, components of extracellular structures, also tend to be highly expressed. On the other hand, transporters are rarely highly expressed. The high expression of MmcX and Archaeal pilin is due, in our opinion, to their involvement in energy metabolism and attachment to minerals.

Lines 185-186: is this also upregulated in these cells? It should be, if one is to consider the stoichiometry of the WL pathway electrons and energy (carbon) substrates.

The SbtA gene was not highly expressed (gene expression level 0.52 (median 0.94)). The CODH related genes were relatively highly expressed (0.9-1.31).

Lines 187-188: This is misleading. They pull the solubility of carbonate into solution by consuming the vanishingly small amount of bicarbonate in solution. Also, carbonates are highly restricted to the near surface portion of ophiolites where higher reduction potentials are commonly encountered. The same could be true for the Cedars. Does this change this interpretation?

As you mentioned, we see a lot of calcites at the surface and as we have never drilled The Cedars serpentinized site, we don't know much about the carbonate distribution in the deep subsurface. However, for example, in the initial report of Oman Drilling Project Hole BT1B (Kelemen et al (2022) JGR <https://doi.org/10.1029/2021JB022352>)) documented that 'Listvenites formed at less than 200°C and (poorly constrained) depths of 25–40 km by reaction with CO₂-rich, aqueous fluids migrating from greater depths, derived from devolatilization of subducting sediments analogous to clastic sediments in the Hawasina Formation, at 400°–500°.', and as for Hawasina Formation, they documented that 'the base of the metamorphic sole is truncated by a fault contact with autochthonous, low-grade metasediments of the Hawasina Formation, composed of pelagic clastic units interlayered with limestones., which indicated that there are carbonate-bearing sedimentary rocks further down the metamorphic sole below the listvenite, which could be the source of CO₂. This is a case of Oman Ophiolite, but CO₂ could be delivered from the deep at The Cedars as well because Franciscan subduction complex is also known to have a complexed geological structure in subsurface.

In the revised manuscript, we added 'carbon acquisition from vanishingly small amount of bicarbonate in solution or perhaps solid phase carbonates involving sodium symport is a widespread strategy for carbon uptake in such a DIC-limited serpentinized setting.' (Line 205-206)

Per the model in Fig. 4, it seems to suggest that RNF is running in reverse (to reduce Fd and NAD) which would drive ion transport into the cell. It should not be depicted as being reversible if this is truly how the authors think the polyheme cytochromes are operating (electron extraction). This is stated by the authors as such in lines 207-209. This makes me confused as to how the ion gradient to power ATP synthesis via the encoded ATPase is formed.

We agree with you. We changed it in the Figure 4 (Now Figure 5c).

Given that Holmes et al. (Mbio 2021, <https://doi.org/10.1128/mbio.02344-21>) suggested that RnfA transports sodium ions into the cell when *M. acetivorans* performs electron fueled methanogenesis coupled with *G. metallireducens* (direct interspecies electron transfer), we showed both possibilities in the figure 4. However, we modified it to display only the direction of electron export from the cell, given the significant difference in the potential of electrons available for donation between the Cedars ecosystem (over -500 mV) and the syntroph of *M. acetivorans*-*G. metallireducens* (~ -250 mV).

Lines 81-252: There is a lot of speculation and interpretation in the results section that might be easier to synthesize by moving to the discussion.

In the Results section, we intended to provide a concise explanation of the current knowledge and the experimental objectives. Upon careful consideration, certain portions were relocated to the Discussion section.

Line 249-250: You have not prepared the reader yet to understand why loss of bifurcating capabilities might be an adaptation to hyper reduced environments.

We agree with you. Microbes may not need the bifurcation capabilities in the hyper reduced environments for the carbon fixation via acetyl-CoA pathway, but this cannot be the reason to lose them. We changed the description.

Lines 251-252: These are not found in deeply rooted archaeal methanogens; not sure about the deeply rooted acetogens.

Figure 3A and Supplementary Figure 3 showed the result of the MmcX-like protein distribution in Halobacteriota, and recent publication indicated that archaeal ECN (extracellular cytochrome nanowires) is widely distributed in bacteria and archaea. (Figures below are from ref Cell 186, 13, 2853 - 2864.e8 (2023)).

Line 264: Which lineage, Archaea? There have been several reports of putative archaeal acetogens from hypersaline environments recently.

The lineage includes orders in the phylum Halobacteriota (Methanocellales, Syntropharchaeales, Methanotrichales and Methanosarcinales).

Line 280: While I appreciate the heterologous biochemical work conducted herein, I don't think it demonstrates that this is what is happening in Met12 to the level of confidence indicated in this sentence.

We changed the sentence to avoid definitive expression (Line 307-313).

Lines 281-282: The use of CO₂ was not demonstrated either, nor was EET. Just in the heterologous expression work.

We changed the sentence to avoid definitive expression (Line 307-313).

Line 294: highly reduced minerals

Corrected as suggested.

Line 287-298: I don't think this work has relevance to what primitive methanogens were doing given the aforementioned comments about this being a clear example of gene loss leading to this putative phenotype. Further, deeply rooted methanogens do not encode cytochromes, making it a stretch to link these observations to early evolving members of this lineage.

As discussed above, we think that the study of metabolic strategies of Met12 could be relevant to the early carbon fixation. However, according to your comments, we deemphasize the description (Line 315-327)

Reviewer #1 (Remarks to the Author):

The authors have adequately addressed all my original concerns, and the authors have made considerable effort to improve their manuscript. In my opinion, this manuscript deserves rapid publication, and will be of great interest to the broad readership of Nature Communications.

Reviewer #3 (Remarks to the Author):

I appreciate the authors many corrections. However, I remain concerned that the authors are misrepresenting their results in the context of what is known of serpentinites, methanogens/acetogens, and the co-evolution of the habitats and their inhabitants. The authors are correct that many studies have suggested serpentinites to be analogs for early Earth conditions. Does this then mean that the aerobes that inhabit modern serpentinites reflect conditions on early Earth? No, yet the authors attempt to rebutte the suggestion that the organisms that they study (Methanocellales) are ancient using the same logic. They are not. Nor is electron bifurcation or electron coupled ion translocation - both are more recent evolutionary innovations. Just like cytochromes - and none of these are primordial.

Thus, while I am excited by the data that is presented and how many of the corrections made further improve the paper, I remain highly skeptical of the stated implications of the work that Met12 has relevance to primordial metabolism. Perhaps the authors could more generalize their findings. For example, cytochromes and the mechanisms that the authors identify for Met12 are not the only mechanisms of EET. And certainly more primitive methanogens and acetogens can use non-cytochrome based methods for EET. Thus, your data point to a role for EET as a potential mechanism to drive early autotrophy via reduced ferredoxin prior to FBEB and Eha/Ehb/Rnf type mechanisms evolving. But do not say that this is one of such mechanisms since existing data do not support this.

REVIEWERS' COMMENTS

Reviewer #1 (Remarks to the Author):

The authors have adequately addressed all my original concerns, and the authors have made considerable effort to improve their manuscript. In my opinion, this manuscript deserves rapid publication, and will be of great interest to the broad readership of Nature Communications.

I appreciate the comments you provided throughout the entire review process. The manuscript has improved significantly because of your input.

Reviewer #3 (Remarks to the Author):

I appreciate the authors many corrections. However, I remain concerned that the authors are misrepresenting their results in the context of what is known of serpentinites, methanogens/acetogens, and the co-evolution of the habitats and their inhabitants. The authors are correct that many studies have suggested serpentinites to be analogs for early Earth conditions. Does this then mean that the aerobes that inhabit modern serpentinites reflect conditions on early Earth? No, yet the authors attempt to rebut the suggestion that the organisms that they study (Methanocellales) are ancient using the same logic. They are not. Nor is electron bifurcation or electron coupled ion translocation - both are more recent evolutionary innovations. Just like cytochromes - and none of these are primordial.

Thus, while I am excited by the data that is presented and how many of the corrections made further improve the paper, I remain highly skeptical of the stated implications of the work that Met12 has relevance to primordial metabolism. Perhaps the authors could more generalize their findings. For example, cytochromes and the mechanisms that the authors identify for Met12 are not the only mechanisms of EET. And certainly more primitive methanogens and acetogens can use non-cytochrome based methods for EET. Thus, your data point to a role for EET as a potential mechanism to drive early autotrophy via reduced ferredoxin prior to FBEB and Eha/Ehb/Rnf type mechanisms evolving. But do not say that this is one of such mechanisms since existing data do not support this.

I appreciate the comments. As you suggested, descriptions related to primordial life were removed or deemphasized in the discussion.

I would like to clarify that aerobes are not present in the deep groundwater at The Cedars. Aerobes are always present at the interface region of this setting. I still think that deep groundwater serves as an analogue of early Earth, as there is no oxygen and almost no organic carbons other than alkane from abiotic reactions. Additionally, I believe it is true that we observe microbes living in this early Earth analogue site, but I agree that the molecular mechanisms (the involved genes) do not necessarily reflect the early evolution of life (at least, we cannot say that based on our data).